# CASK and FARP localize two classes of post-synaptic ACh receptors thereby promoting cholinergic transmission

Lei Li[1]◉, Haowen Liu[1]◉, Kang-Ying Qian[2]◉, Stephen Nurrish[3,4]◉, Xian-Ting Zeng[2], Wan-Xin Zeng[2], Jiafan Wang[1], Joshua M. Kaplan[3,4,5]*, Xia-Jing Tong[2]*, Zhitao Hu[1]*

**1** Queensland Brain Institute, Clem Jones Centre for Ageing Dementia Research (CJCADR), The University of Queensland, Brisbane, Australia, **2** School of Life Science and Technology, ShanghaiTech University, Shanghai, China, **3** Department of Molecular Biology, Massachusetts General Hospital, Boston, Massachusetts, United States of America, **4** Department of Neurobiology, Harvard Medical School, Boston, Massachusetts, United States of America, **5** Program in Neuroscience, Harvard Medical School, Boston, Massachusetts, United States of America

◉ These authors contributed equally to this work.
* kaplan@molbio.mgh.harvard.edu (JMK); tongxj@shanghaitech.edu.cn (X-JT); z.hu1@uq.edu.au (ZH)

## Abstract

Changes in neurotransmitter receptor abundance at post-synaptic elements play a pivotal role in regulating synaptic strength. For this reason, there is significant interest in identifying and characterizing the scaffolds required for receptor localization at different synapses. Here we analyze the role of two *C. elegans* post-synaptic scaffolding proteins (LIN-2/CASK and FRM-3/FARP) at cholinergic neuromuscular junctions. Constitutive knockouts or muscle specific inactivation of *lin-2* and *frm-3* dramatically reduced spontaneous and evoked post-synaptic currents. These synaptic defects resulted from the decreased abundance of two classes of post-synaptic ionotropic acetylcholine receptors (ACR-16/CHRNA7 and levamisole-activated AChRs). LIN-2's AChR scaffolding function is mediated by its SH3 and PDZ domains, which interact with AChRs and FRM-3/FARP, respectively. Thus, our findings show that post-synaptic LIN-2/FRM-3 complexes promote cholinergic synaptic transmission by recruiting AChRs to post-synaptic elements.

## Author summary

Post-synaptic receptors are immobilized to form clusters at synapses. This process requires the participation of scaffolding proteins, and is thought to be of critical importance in central neurotransmission, ensuring the synaptic efficiency and fidelity. Here we report that two scaffold proteins, LIN-2/CASK and FRM-3/FARP, function as major synaptic organizers at *C. elegans* cholinergic neuromuscular junctions. LIN-2/CASK belongs to the MAGUK protein family and has been found to be important for cellular junctions. Knockout of *lin-2* or *frm-3* leads to severe defects in cholinergic synaptic transmission, with the mEPSC (<u>m</u>iniature <u>e</u>xcitatory <u>p</u>ost<u>s</u>ynaptic <u>c</u>urrent) amplitude and synaptic localization of ionotropic acetylcholine receptors (AChRs) being dramatically reduced in

**Data Availability Statement:** Data generated or analysed during this study are included in this published article or available upon request.

**Funding:** This work was supported by grants from the NHMRC (GNT1122351 to Z.H.), BBRF (NARSAD Young Investigator grant 24980 to Z.H.), NIH (NS32196 to J.M.K.), National Key Research and Development Program of China (2021ZD0202500 to X-J. T.), Basic Research Project from the Science and Technology Commission of Shanghai Municipality (19JC1414100 to X-J.T.), and National Natural Science Foundation of China (32170963 to X-J. T.). The funders had no role in study design, data collection and analysis, decision to publish, or preparation of the manuscript.

**Competing interests:** The authors have declared that no competing interests exist.

*lin-2* and *frm-3* mutants. Conditional knockout of *lin-2* or *frm-3* in body wall muscles produced all phenotypes observed in the constitutive mutants, demonstrating that LIN-2/CASK and FRM-3/FARP function postsynaptically, regulating synaptic localization of AChRs. The function of LIN-2/CASK requires its SH3 and PDZ domains, which mediate interaction with AChRs and FRM-3. Collectively, our results have shown that scaffolding proteins LIN-2/CASK and FRM-3/FARP regulate postsynaptic AChRs, thereby promoting cholinergic synaptic transmission.

## Introduction

Synaptic connections mediate fast intercellular communication in the nervous system. Synapses are elaborate structures comprising hundreds of pre- and post-synaptic proteins, each highly concentrated at the sites of intercellular contact. Although synaptic proteins are highly enriched at these contacts, they nonetheless must retain the ability to undergo dynamic processes (e.g. insertion, removal, intermolecular contacts, and post-translational modifications). The abundance of a protein at a synapse often dictates signaling properties of that synapse. For example, the location and abundance of the synaptic vesicle priming protein (UNC-13) controls the probability and kinetics of release [1,2]. Similarly, the abundance of post-synaptic receptors dictates the size of synaptic currents [3,4]. Beyond recruiting rate limiting synaptic components, synaptic transmission also relies upon the precise coordination of pre- and post-synaptic specializations. For example, the site of synaptic vesicle (SV) fusion in presynaptic cells (termed Active Zones) must be precisely aligned with clustered receptors in the contacting post-synaptic cell. Slight shifts in trans-synaptic alignment results in significant alterations in the ongoing signaling at that synapse [5–7]. For these reasons, there is significant interest in determining how rate limiting synaptic components are localized and coordinated trans-synaptically.

To become functional, assembled ionotropic receptors must be transported to the plasma membrane and clustered at post-synaptic elements, where they are immobilized by synaptic scaffolding proteins. In rodents, several scaffolding proteins have been identified for localizing ionotropic GABA (neuroligin 2, collybistin, gephyrin) [8–10], glutamate (PSD95, neuroligin 1 and 3) [11–13], and ACh (rapsyn, MuSK) receptors [14,15]. Analysis of invertebrate synapses have identified additional synaptic scaffolds. In *Drosophila*, ionotropic glutamate receptors are immobilized at neuromuscular junctions by CASK, Coracle/Band 4.1 proteins, Filamin, and Discs-Large (DLG) [16–19]. At *C. elegans* NMJs, ionotropic GABA receptors are localized by Neuroligin, FRM-3/FARP, LIN-2/CASK, and UNC-40/DCC [20–23]. Interestingly, a recent study suggests that *C. elegans* ionotropic ACh receptors (AChRs) are synaptically localized by an overlapping set of scaffolds (FRM-3/FARP, LIN-2/CASK, and UNC-40/DCC) [24]; however, this study did not analyze how cholinergic transmission and behavior were altered in these scaffold mutants.

CASK (a MAGUK family scaffold protein) regulates synaptic function [18,25–27]. Apart from the PDZ, SH3, and GK domains found in all MAGUK proteins, CASK also contains an N-terminal $Ca^{2+}$/calmodulin-dependent protein kinase (CaM) domain (Fig 1A). CASK orthologs are found in invertebrates and vertebrates, suggesting that CASK's synaptic function is strongly conserved across phylogeny. In mammals, CASK is primarily expressed in the brain and localizes in synaptic regions [27–29]. Similar to other scaffolds, CASK participates in multiple interactions in different signalling processes, and has been implicated in synaptic protein targeting, synaptic organization, and transcriptional regulation [29,30]. Human CASK has also

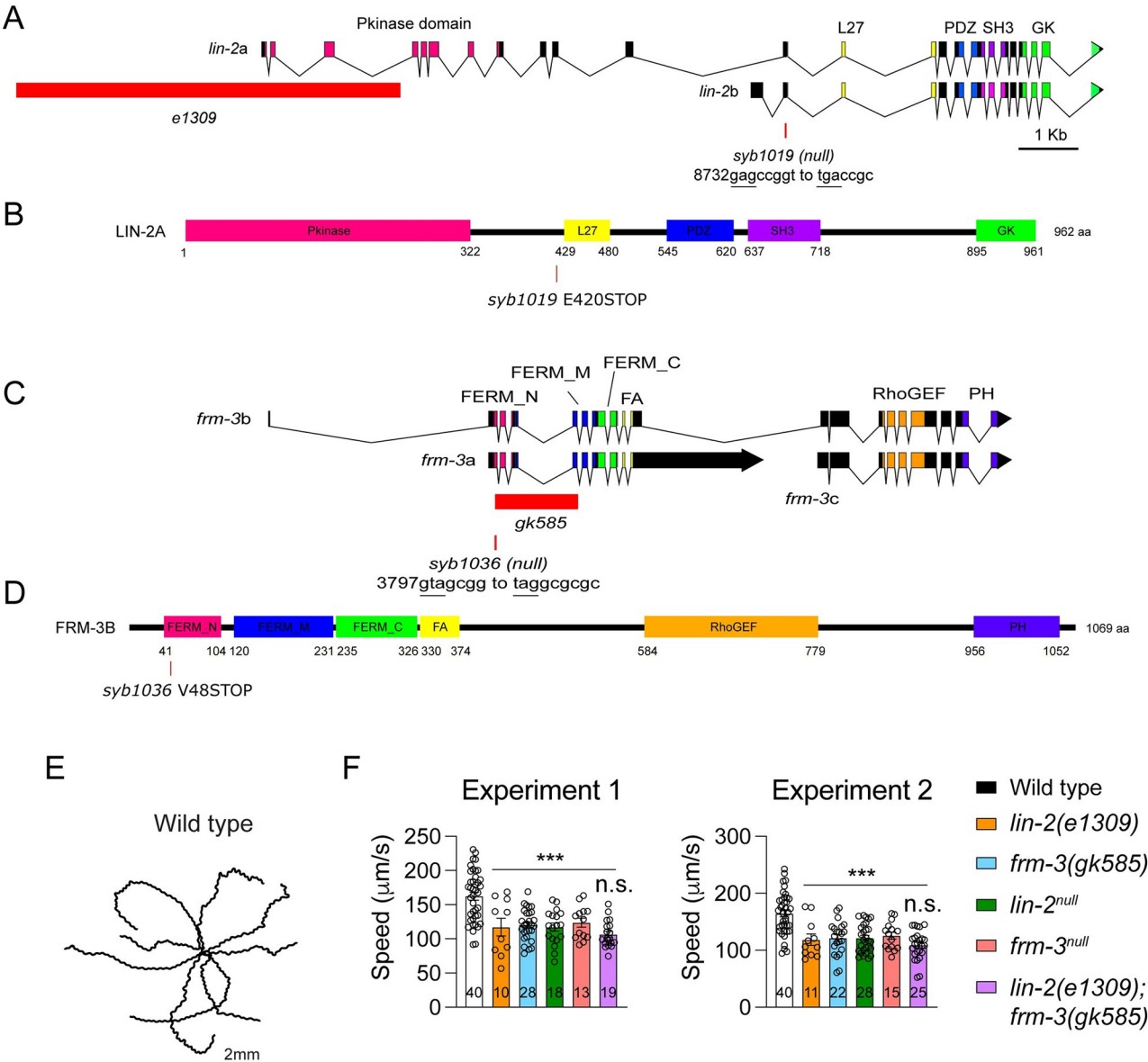

**Fig 1. LIN-2 and FRM-3 are required to maintain locomotory behaviour.** (A) Genomic structure of *lin-2* and *frm-3* locus, mutant allele characterization, and protein architecture for LIN-2 and FRM-3. The whole promoter region of *lin-2a* and part of the kinase domain of LIN-2A is deleted in the *e1309* mutants. The FERM domain in both *frm-3a* and *frm-3b* is deleted in the *gk585* mutants. *Syb1019* and *syb1036* are stop codons in *lin-2* and *frm-3* that inactivate the expression of *lin-2a* and *lin-2b*, and *frm-3a* and *frm-3b*. (E, F) Locomotion speed is reduced by the loss of LIN-2 and FRM-3. Representative trajectories of locomotion in wild type (E) and mean locomotion speed in wild type, *lin-2(e1309)*, *frm-3(gk585)*, *lin-2(syb1019)*, *frm-3(syb1036)*, and *lin-2(e1309);frm-3(gk585)* mutants. To measure locomotion speed, young adult animals were washed with a drop of PBS and then transferred to fresh NGM plates with no bacterial lawn (30 worms per plate). Worm movement recordings (under room temperature 22°C) were started 10 min after the worms were transferred. A 2 min digital video of each plate was captured at 3.75 Hz frame rate by WormLab System (MBF Bioscience). Average speed and tracks were generated for each animal using WormLab software. To confirm the repeatability of the data, the locomotion speed was measured in two independent experiments in two days. For each mutant, around 10–40 animals were analyzed in one experiment. Significance was tested for each experiment. Data are mean ± SEM (**, $p < 0.01$, ***, $p < 0.001$ when compared to wild type; n.s., non-significant; one-way ANOVA). The number of worms analyzed for each genotype is indicated in the bar.

been linked to neurological disorders such as autism and brain malformation [31,32]. In mouse cortical neurons and *Drosophila* NMJ, glutamatergic synaptic transmission is impaired by the knockout of CASK [18,33]. In *C. elegans*, LIN-2/CASK together with its binding partner FRM-3 (ortholog of the mammalian scaffold FARP), promote GABAergic synaptic transmission by immobilizing GABA$_A$ receptors at the NMJ [20, 21]. A recent study showed that LIN-2/CASK and FRM-3/FARP are also required for stabilizing ACR-16/CHNRA7 receptors at the *C. elegans* NMJ [24]. Despite these advances in both vertebrates and invertebrates, key questions remain to be addressed regarding the function of CASK in cholinergic synapses. How do CASK and its binding partners regulate AChRs? Does CASK function by similar mechanisms at cholinergic and GABAergic synapses?

Here we show that LIN-2/CASK and FRM-3/FARP promote cholinergic transmission by recruiting two classes of ionotropic AChRs, ACR-16/CHRNA7 and levamisole-activated AChRs (L-AChRs), to post-synaptic elements. The SH3 domain in LIN-2/CASK plays important roles by directly interacting with ionotropic AChRs. Our results provide a detailed picture of how LIN-2/FRM-3 complexes shape the function of cholinergic synapses.

## Results

### Loss of function of *lin-2* or *frm-3* decreases locomotion speed

To assess LIN-2 and FRM-3's impact on behavior, we measured locomotion speed in *lin-2* and *frm-3* mutants. The *lin-2* gene encodes two isoforms, LIN-2A and LIN-2B, which share the PDZ, SH3, and GK domains while only LIN-2A contains the CaMK homology domain [34]. The *lin-2(e1309)* mutation deletes an N-terminal region of LIN-2A but has no effect on LIN-2B (Fig 1A). The *frm-3* gene encodes three isoforms, two of which (FRM-3A and B) possess a FERM domain. The *gk585* allele is a deletion disrupting both *frm-3*a and b (Fig 1C). Locomotion speed was significantly decreased in *lin-2(e1309)* mutants and *frm-3(gk585)* mutants (Fig 1E and 1F), suggesting that LIN-2A and FRM-3 are required for the function of the locomotion circuit. No additional decrease in locomotion speed was observed in *lin-2 (e1309)*; *frm-3(gk585)* double mutants, indicating that LIN-2 and FRM-3 act together to promote locomotion.

Because the *lin-2(e1309)* and *frm-3(gk585)* may not be null mutants, we used CRISPR to introduce early nonsense mutations in *lin-2(syb1019)* and *frm-3(syb1036)*. The *lin-2(syb1019)* nonsense mutation is in an exon shared by LIN-2A and B (Fig 1A and 1B); consequently, both isoforms are inactivated by this allele. The *lin-2(syb1019)* mutants exhibited similar vulva development and egg-laying defects to those seen in *lin-2(e1309)* mutants. Similarly, the *frm-3 (syb1036)* nonsense mutation is in an exon shared by the FRM-3A and B isoforms (Fig 1C and 1D). Hereafter, we refer to *syb1019* and *syb1036* as *lin-2* and *frm-3* null mutants respectively. The locomotion speed defects in *lin-2* and *frm-3* null mutants were similar to those in *lin-2 (e1309)* and *frm-3(gk585)* mutants (Fig 1F). Together, these results reveal that LIN-2/CASK and FRM-3/FARP play important roles in locomotion behavior.

### Cholinergic synaptic transmission is severely impaired in *lin-2* and *frm-3* mutants

We next asked if LIN-2/CASK and FRM-3/FARP are required for cholinergic synaptic transmission. *C. elegans* body wall muscles receive synaptic inputs from both cholinergic and GABAergic motor neurons [35]. Prior studies demonstrated that the synaptic abundance and mobility of UNC-49/GABA$_A$ receptors (GABA$_A$R) at the NMJs are significantly disrupted in *lin-2(e1309)* and *frm-3(gk585)* mutants, leading to decreased amplitude of the miniature

inhibitory postsynaptic currents (mIPSCs) [20,21]. By recording the miniature and stimulus-evoked excitatory postsynaptic currents (mEPSCs and evoked EPSCs) in *lin-2(e1309)* and *frm-3(gk585)* mutants, we found that the cholinergic excitatory synaptic transmission was also significantly impaired, with the mEPSC amplitude, and evoked EPSC amplitude and charge transfer being markedly decreased (Fig 2A, 2B, 2E and 2F). The mEPSC frequencies were also significantly decreased in *lin-2(e1309)* and *frm-3(gk585)* mutants (60% and 30% of WT; Fig 2B). The *lin-2(e1309)*; *frm-3 (gk585)* double mutants exhibited similar synaptic transmission defects to those found in *frm-3(gk585)* single mutants, including mEPSCs and evoked EPSCs, indicating that LIN-2/CASK and FRM-3/FARP function together at the ACh NMJs.

Compared to *lin-2(e1309)* mutants, *lin-2* null mutants exhibited more severe defects in cholinergic synaptic transmission, including larger decreases in mEPSC frequency and amplitude, and evoked EPSC amplitude and charge transfer (Fig 2A, 2B, 2E and 2F). The mEPSC frequency in *frm-3* null mutants was also significantly lower than in *frm-3(gk585)* mutants. Moreover, mIPSC frequencies, which were unaltered in *lin-2(e1309)* and *frm-3(gk585)* mutants, were significantly decreased in both *lin-2* and *frm-3* null mutants (Fig 2C and 2D). These results demonstrate that synaptic transmission was more severely impaired in the *lin-2* and *frm-3* null mutants, further establishing that these scaffolds play a pivotal role in the function of cholinergic synapses. It should be noted that LIN-2's impact on cholinergic transmission does not require its other binding partners LIN-7/Velis and LIN-10/Mint [25,36], as the *lin-7* and *lin-10* mutants exhibited normal mEPSCs and evoked EPSCs (S1 Fig).

The decreased mEPSC amplitude suggests that postsynaptic ionotropic AChRs are disrupted in *lin-2* and *frm-3* mutants (e.g., decrease in synaptic abundance). It should be noted that smaller mEPSC amplitudes may cause some mEPSC events to become undetectable, thereby reducing mEPSC frequency. This is supported by the fact that mEPSC frequency is decreased by 55% in *acr-16* mutants, which lack nicotine-sensitive receptors that are orthologous to mammalian CHRNA7 receptors (S2A and S2B Fig). Thus, the decreased mEPSC rates observed in *lin-2* and *frm-3* mutants could be caused by the decreased mEPSC amplitude. Moreover, the severe defects in mEPSCs but normal locomotion speed in the *acr-16* mutants (S2C Fig) also explained the moderate decrease in locomotion in *lin-2* and *frm-3* mutants. Taken together, our results revealed that LIN-2/CASK and FRM-3/FARP play essential roles in cholinergic synaptic transmission, likely by promoting post-synaptic responses to ACh.

## Muscle specific knockouts of *lin-2* and *frm-3* cause cholinergic transmission defects

LIN-2/CASK and FRM-3/FARP are expressed in both motor neurons and postsynaptic body wall muscles [20,21]. To determine which cells require LIN-2/CASK and FRM-3/FARP function, we generated *lin-2(nu743 FLEX)* and *frm-3(nu751 FLEX)* alleles that allow inactivation of these genes by the CRE recombinase. Using CRISPR, we introduced a stop cassette into *lin-2* and *frm-3* introns (in the opposite orientation). For both genes, the STOP cassette was inserted into an intron shared by both isoforms (Fig 3A). The STOP cassette is bounded by FLEX recombination signals mediating CRE-induced inversions. In this manner, CRE expression inverts the stop cassette thereby blocking expression of LIN-2(A and B) and FRM-3(A and B). Neuron-specific knockouts (KO) were made by expressing P*sbt-1*::Cre (termed Neuron[Cre]) while body wall muscle KOs were made by expressing P*myo-3*::Cre (termed Muscle[Cre]).

We next examined synaptic transmission following *lin-2* and *frm-3* muscle and neuron KO. Compared to control *lin-2(nu743)* and *frm-3(nu751)* mutants (lacking Cre expression), the frequency and amplitude of mEPSCs, and the amplitude and charge transfer of evoked EPSCs, were all severely reduced by Muscle[Cre], but were not changed by Neuron[Cre] (Fig 3B–3I),

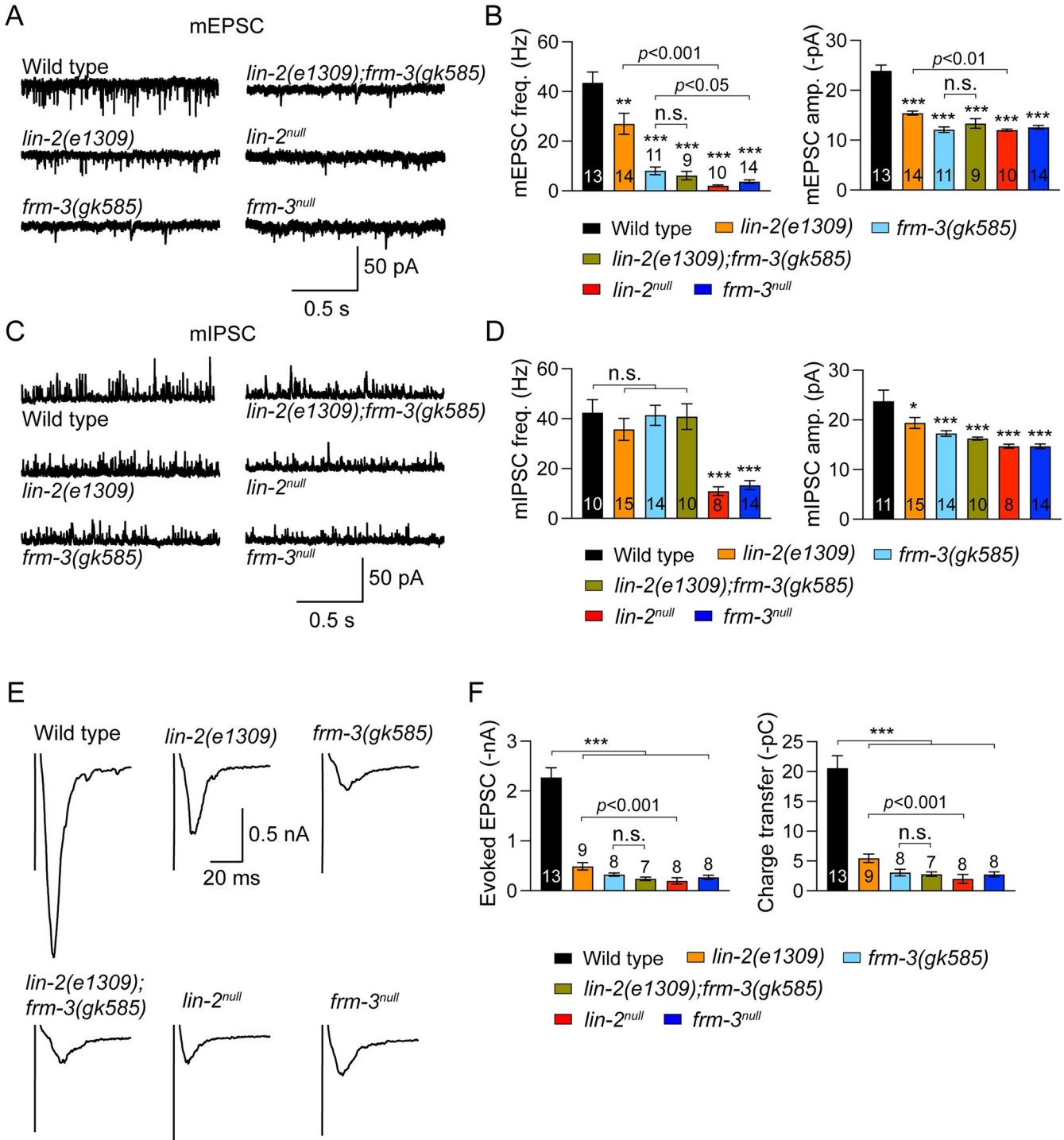

**Fig 2. Cholinergic synaptic transmission is dramatically decreased in *lin-2* and *frm-3* mutants.** (A, B) Representative traces of mEPSCs and quantification of mEPSC frequency and amplitude from the indicated genotypes, including wild type, *lin-2(e1091)*, *frm-3(gk585)*, *lin-2(e1091);frm-3(gk585)*, *lin-2^null^*, and *frm-3^null^* mutants. (C, D) Example traces of mIPSCs and summary of mIPSC frequency and amplitude from the same genotypes in A. (E, F) Evoked EPSC traces and quantification of EPSC amplitude and charge transfer. Data are mean ± SEM ($*$, $p < 0.05$, $***$, $p < 0.001$ when compared to wild type; n.s., non-significant; one-way ANOVA). The number of worms analyzed for each genotype is indicated in the bar.

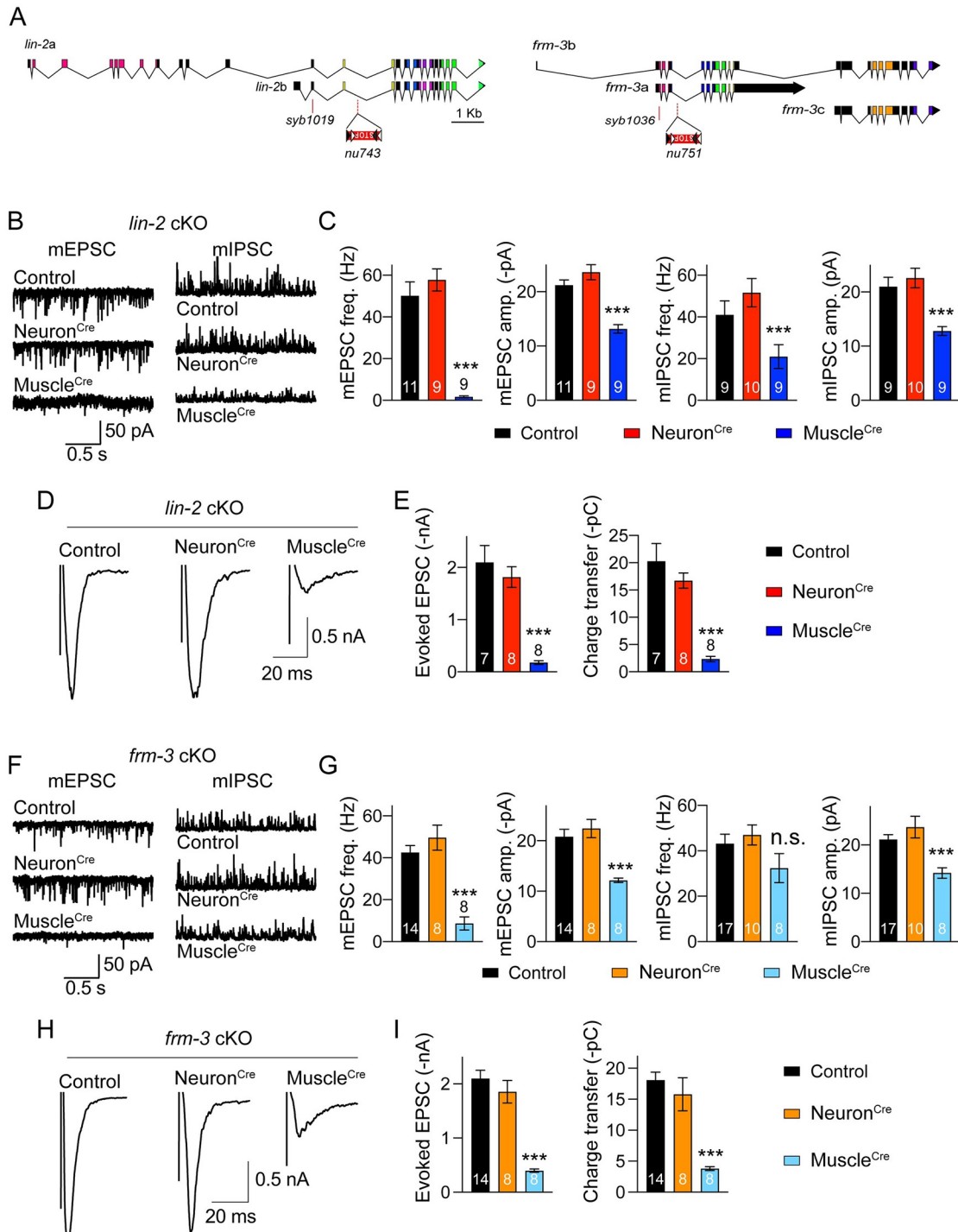

**Fig 3. Conditional knockouts of *lin-2* and *frm-3* in muscle reduce cholinergic transmission.** (A) The STOP cassette is inserted into the common intron region in *lin-2a* and *lin-2b*, and in *frm-3a* and *frm-3b*. The presence of CRE inverts the STOP cassette to inactivate expression of *lin-2* and *frm-3* isoforms. (B) Example traces of mEPSCs from control (no expression of CRE), neuron KO (expressing CRE in all neurons driven by the *sbt-1* promoter, Neuron^Cre), and muscle KO (expressing CRE in body wall muscles driven by the *myo-3* promoter Muscle^Cre) of *lin-2*. (C) Quantification of mEPSC frequency, amplitude, decay, and charge in B. (D) Evoked EPSC traces from control, *lin-2* neuron KO and muscle KO. (E) Quantitation of evoked EPSC amplitude, and charge transfer. (F, G) mEPSC traces and averaged mEPSC frequency, amplitude, decay and charge from control, *frm-3* neuron KO, and *frm-3* muscle KO. (H, I) Evoked EPSCs and summary of EPSC amplitude and charge transfer from control and *frm-3* conditional KO mutants. Data are mean ± SEM (**, $p < 0.01$, ***, $p < 0.001$ when compared to control; one-way ANOVA). The number of worms analyzed for each genotype is indicated in the bar.

demonstrating that LIN-2/CASK and FRM-3/FARP both promote cholinergic synaptic transmission by acting in body wall muscles. Similarly, Muscle[Cre] also significantly decreased the mIPSC amplitude in *lin-2(nu743)* and *frm-3(nu751)* mutants (Fig 3B, 3C, 3F and 3G), indicating that LIN-2 and FRM-3 also act in muscle to regulate postsynaptic GABA$_A$ receptors, consistent with prior studies [20,21]. However, unlike cholinergic synapses, Muscle[Cre] decreased the mIPSC frequency by only 50% in *lin-2(nu743)* mutants (versus 90% in *lin-2* null mutants), and did not change mIPSC frequency in *frm-3(nu751)* mutants. Collectively, these results indicate that LIN-2 and FRM-3 control synaptic transmission at both cholinergic and GABAergic synapses but that the detailed mechanisms likely differ at these two synapses.

We next analyzed the locomotion speed in *lin-2* and *frm-3* conditional knockout mutants. Our results showed that the speed is significantly decreased in *lin-2* Muscle[Cre] mutants (S3 Fig), consistent with that in *lin-2* null mutants (Fig 1F). However, the speed is not altered in either the Muscle[Cre] or Neuron[Cre] mutants of *frm-3*. One possibility is that the efficiency of the Cre recombinase in *frm-3(nu751)* is not as strong as that in *lin-2(nu743)*. This is likely because the mIPSC frequency is not decreased in *frm-3* Muscle[Cre] mutants, unlike *frm-3* null mutants (Figs 1F and 2D).

Notably, the mEPSCs and evoked EPSCs in the Muscle[Cre] mutants are comparable with those in the *lin-2* and *frm-3* null mutants (Fig 2), indicating that the cholinergic synaptic transmission defects result from the loss of LIN-2 and FRM-3 postsynaptic function.

The unchanged synaptic transmission in the Neuron[Cre] mutants of *lin-2* and *frm-3* may arise from a failure of the Cre recombinase to inactivate *lin-2* and *frm-3* in ACh neurons. To address this concern, we asked if Cre recombinase expression could inactivate an essential presynaptic gene, *unc-2*. The *unc-2* gene encodes the P/Q-type Ca$^{2+}$ channel in *C. elegans*. Loss of *unc-2* function leads to strong decrease in both locomotion rate and Ca$^{2+}$-dependent neurotransmitter release [37,38]. The *unc-2(nu657* FLEX) allele contains an inverted STOP cassette in intron 17. The presence of CRE is expected to inactivate *unc-2*, leading to behaviour and transmission defects. As shown in S4 Fig, Neuron[Cre] in *unc-2 (nu657)* significantly decreased locomotion speed, mEPSC frequency, and evoked EPSC amplitude and charge transfer. These results are similar to those observed in *unc-2* mutants, demonstrating that the Cre recombinase is highly efficient in cholinergic motor neurons.

## LIN-2/CASK and FRM-3/FARP regulate the synaptic abundance of postsynaptic AChRs

The dramatic decrease in mEPSC amplitude in *lin-2* and *frm-3* null mutants suggests that the ionotropic AChRs are impaired in postsynaptic body wall muscles. There are two classes of ionotropic AChRs expressed on body wall muscles in *C. elegans*, nicotine-sensitive homo-pentameric receptors (nAChRs) consisting of five ACR-16/CHRNA7 subunits, and levamisole-sensitive hetero-pentameric receptors (L-AChRs) containing alternative α subunits (UNC-38, UNC-63, and LEV-8) and non-α subunits (UNC-29 and LEV-1) [39,40]. These two types of AChRs can be distinguished by their differential desensitization in response to long-time exposure to acetylcholine [41]. Moreover, because of the difference in channel conductance, activation of nAChRs elicits large and rapidly decaying responses, and L-AChRs produce small synaptic responses with slow decay when activated [42–44].

To estimate the synaptic abundance of nAChRs, we imaged ACR-16::RFP (expressed by the *myo-3* promoter in a single copy transgene). ACR-16::RFP exhibits a punctate distribution in the nerve cords, consistent with previous observation [42]. Compared to wild-type animals, the *lin-2* and *frm-3* null mutants exhibited significantly reduced ACR-16::RFP puncta fluorescence in the dorsal nerve cord (Fig 4A and 4B), consistent with results reported in a recent

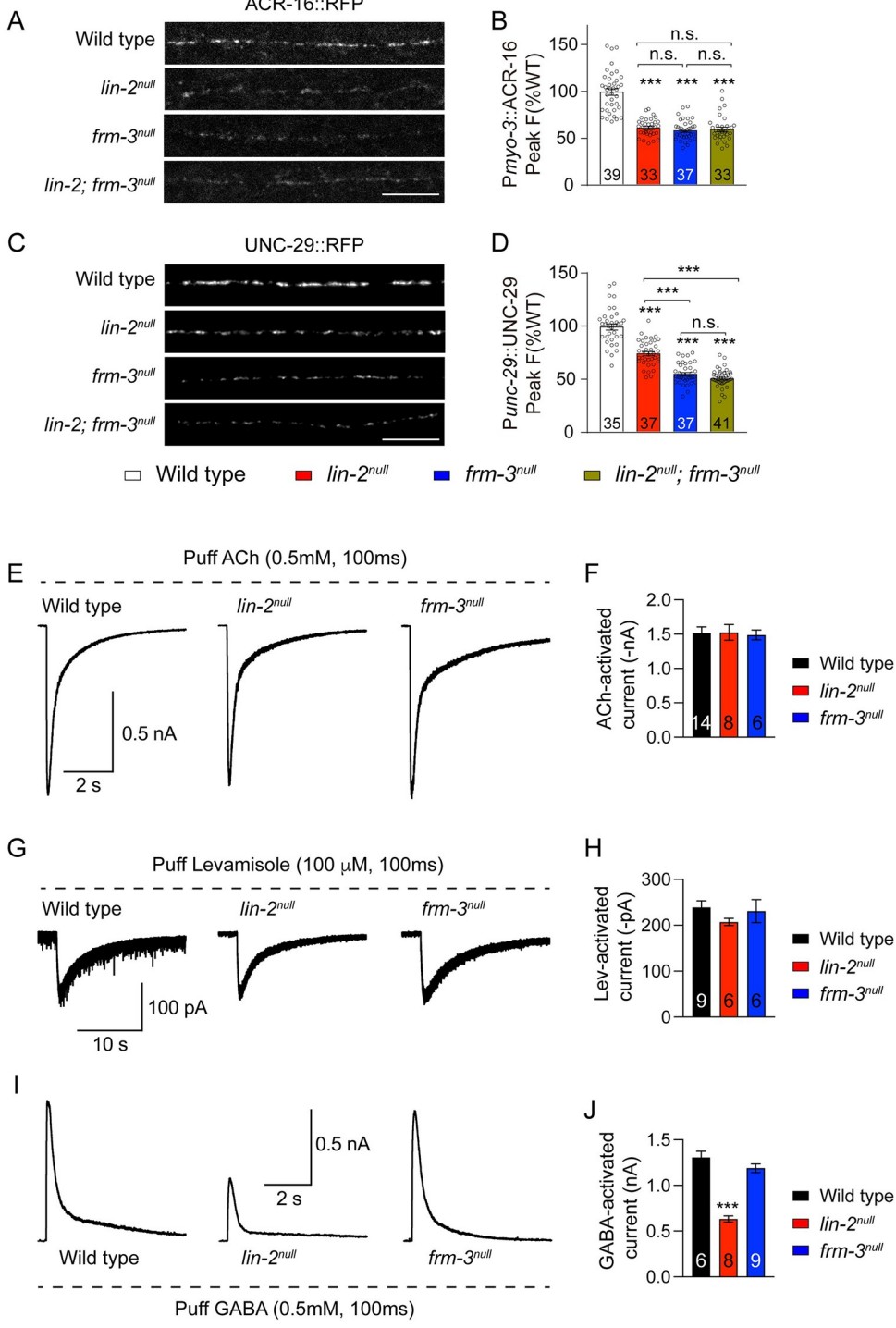

**Fig 4. LIN-2 and FRM-3 regulate the synaptic abundance but not surface expression level of AChRs.** (A-D) ACR-16::RFP and UNC-29::RFP synaptic abundance were decreased in *lin-2^null^* and *frm-3^null^* mutants. Representative images (A, C, scale bar 10 μm) and mean puncta intensity (B, D) are shown. The wild type is normalized to 1. (E-H) ACh- and Levamisole-activated currents were unaltered in *lin-2^null^* and *frm-3^null^* mutants. Representative traces (E, G) and mean current amplitude (F, H) are shown. (I, J) GABA-activated currents were decreased by 50% in *lin-2^null^* mutants but were unchanged in *frm-3^null^* mutants. Data are mean ± SEM (***, $p < 0.001$ when compared to control; one-way ANOVA). The number of worms analyzed for each genotype is indicated in the bar.

study [24]. These observations are consistent with the decreased mEPSC and evoked EPSC amplitude (Fig 2). We next asked if loss of LIN-2 and FRM-3 alters the synaptic abundance of endogenously expressed L-AChRs, by analyzing the puncta intensity produced by the *unc-29 (kr208)* allele, which contains an RFP tag [22]. Our data showed that the UNC-29::RFP puncta fluorescence was also remarkably decreased in both *lin-2* and *frm-3* null mutants (Fig 4C and 4D). There were no additional decreases in ACR-16::RFP and UNC-29::RFP in the *lin-2;frm-3* double mutants (Fig 4B and 4D), suggesting that LIN-2 and FRM-3 function together to regulate both ACR-16/CHRNA7 and L-AChR synaptic abundance.

LIN-2 and FRM-3 regulation of post-synaptic AChRs is also supported by the altered kinetics of mEPSCs. In *lin-2* and *frm-3* null mutants, as well as their Muscle^Cre mutants, mEPSCs exhibited significantly slower decay kinetics (S5 Fig), which likely results from a greater contribution of L-AChRs to the synaptic current. These observations further suggest a defect in post-synaptic AChRs in *lin-2* and *frm-3* mutants. Together, our results demonstrate that LIN-2 and FRM-3 act in the same genetic pathway regulating the synaptic abundance of two classes of ionotropic AChRs in body wall muscles.

To examine the function of endogenous nAChRs and L-AChRs on the muscle surface, we measured acetylcholine- and levamisole-activated synaptic responses. A pulse application of ACh (0.5mM, 100ms) onto body wall muscle elicited a large and robust current in wild-type animals. In *lin-2* and *frm-3* null mutants, ACh-activated currents were indistinguishable from that in wild type (Fig 4E and 4F). Puffing levamisole (100μM, 100ms) produced a small but long-lasting current in wild-type animals, and the levamisole-evoked currents were also unaltered in *lin-2* and *frm-3* null mutants (Fig 4G and 4H). These results indicate that inactivating LIN-2 and FRM-3 did not alter the total expression level or surface delivery of nAChRs and L-AChRs in muscles. Instead, inactivating LIN-2 and FRM-3 prevent nAChRs and L-AChRs clustering or destabilize receptor clusters at synapses, resulting in decreased ACR-16::RFP and UNC-29::RFP puncta fluorescence and decreased mEPSC and evoked EPSC amplitudes.

The impact of LIN-2 and FRM-3 on post-synaptic AChR clustering is similar to their role at GABAergic synapses [20,21]. In *lin-2(e1309)* and *frm-3(gk585)* mutants, UNC-49::GFP/ GABA$_A$ puncta intensity and mIPSC amplitude were significantly decreased while currents evoked by exogenous GABA puffs were unchanged [20]. Because the null mutants of *lin-2* and *frm-3* displayed stronger transmission defects, we re-examined the effects of *lin-2* and *frm-3* knockouts on GABA$_A$R. We found that GABA-activated currents were significantly reduced (by 50%) in *lin-2* null mutants but were unchanged in *frm-3* null mutants (Fig 4I and 4J). These results suggest that LIN-2 also controls total surface GABA$_A$R levels in muscles.

To further confirm the regulation of LIN-2 and FRM-3 on the surface levels of AChRs and GABA$_A$Rs, we also measured ACh-, levamisole-, and GABA-activated currents in the muscle cKO mutants of *lin-2* and *frm-3*. Overall, the results were similar to those observed in the null mutants (S6 Fig). We only found a significant decrease in GABA-activated current in *lin-2* Muscle^Cre mutants, providing further support for a previously unknown role of LIN-2 in the trafficking of GABA$_A$Rs to the cell surface.

## Synaptic morphology is normal in *lin-2* and *frm-3* mutants

Thus far, our results suggest that LIN-2 and FRM-3 are required for post-synaptic function; however, it remains possible that these scaffolds also have important pre-synaptic functions. To address this possibility, we examined the synaptic ultrastructure of cholinergic motor neurons. Morphometric analyses showed that the terminal size and total number of SVs in *lin-2* and *frm-3* mutants were indistinguishable from that in wild-type control (Fig 5A and 5B).

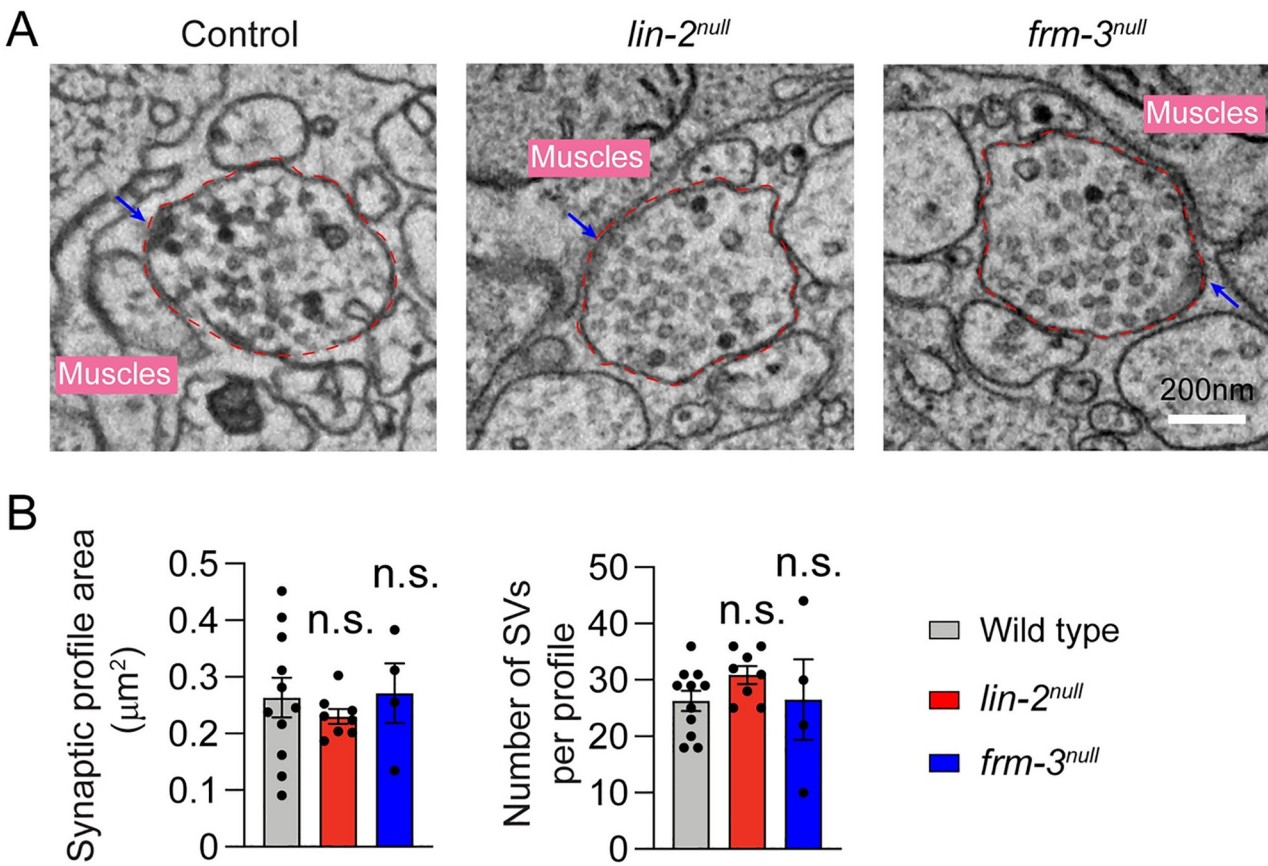

**Fig 5. Ultrastructural analysis of *lin-2* and *frm-3* mutants.** (A) Representative micrographs of wild-type control, *lin-2*$^{null}$ mutant, and *frm-3*$^{null}$ mutant synaptic profiles. Arrows indicate the dense projection. The terminal area is labelled by the dashed red line. Scale bar = 200 nm. (**B**) Quantification of synaptic terminal area (measured in μm$^2$) and total number of SVs. Data are mean ± SEM.

These results indicate that *lin-2* and *frm-3* mutations do not have dramatic effects on the morphology of presynaptic terminals.

## SV abundance is not impaired in *lin-2* and *frm-3* mutants

To further examine the potential effects of LIN-2 and FRM-3 on presynaptic terminals, we examined SV abundance in *lin-2* and *frm-3* mutants. SV abundance was analyzed by imaging synaptic vesicle markers UNC-57::mCherry (expressed by the *unc-129* promoter in a single copy transgene). UNC-57::mCherry fluorescence displays punctate distribution in dorsal nerve cord in wild-type animals (Fig 6A). Our data showed that UNC-57::mCherry in *lin-2* and *frm-3* mutants have comparable puncta fluorescence and density compared to wild-type controls (Fig 6B), indicating that presynaptic SV abundance is normal in these two mutants. These results are consistent with the observation in EM in which the total number of SVs is unaltered in *lin-2* and *frm-3* mutants (Fig 5B). Together, our results indicate that SV abundance is unchanged in *lin-2* and *frm-3* mutants, and consequently that the cholinergic transmission defects observed in these mutants are likely a consequence of post-synaptic defects.

## The SH3 domain in LIN-2/CASK binds to both ACR-16 and UNC-29

The decreased nAChRs and L-AChRs in *lin-2* and *frm-3* mutants indicates that LIN-2/CASK and FRM-3/FARP may physically bind to these receptors and regulate their synaptic

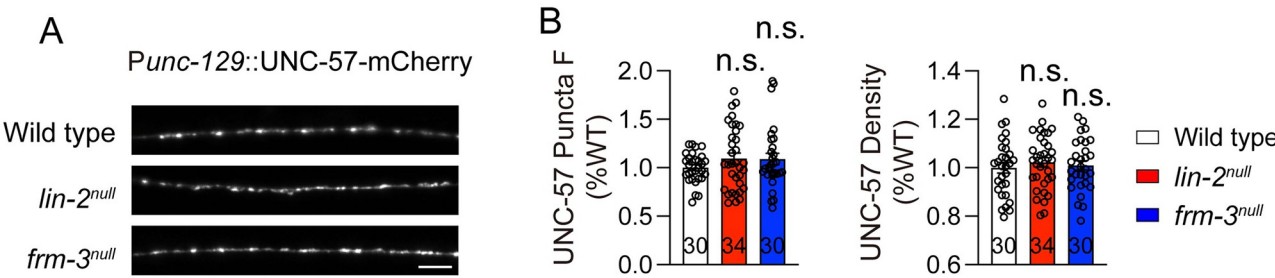

**Fig 6. Presynaptic SV abundance is normal in *lin-2* and *frm-3* mutants.** Expression levels of UNC-57::mCherry was unchanged in *lin-2^null^* and *frm-3^null^* mutants. Representative images (A, scale bar 10 μm) and mean peak puncta intensity and puncta density (B) are shown. The wild type is normalized to 1. Data are mean ± SEM. The number of worms analyzed for each genotype is indicated in the bar.

abundance. To test this, we performed yeast two-hybrid (Y2H) assays. Our data showed that LIN-2A could bind to the second cytoplasmic loop of both ACR-16 and UNC-29 (Fig 7A–7C), with stronger interaction with the ACR-16 loop than the UNC-29 loop. These results are consistent with the findings that *lin-2* or *frm-3* knockout led to a decrease in puncta fluorescence of AChRs (Fig 4). However, no direct interactions were detected between FRM-3 and the cytoplasmic loop of ACR-16 or UNC-29. By contrast, a recent study reported that ACR-16 binds both LIN-2 and FRM-3 [24]. This discrepancy could result from the use of different binding assays (i.e. Y2H versus GST pull-down). Nevertheless, since FRM-3 physically binds to LIN-2, FRM-3's impact on AChRs is likely mediated by LIN-2/FRM-3/ACR-16 and LIN-2/FRM-3/UNC-29 ternary complexes.

LIN-2 contains PDZ and SH3 domains that mediate protein-protein interactions. To examine which domain(s) in LIN-2 mediates the interaction with ACR-16 and UNC-29, we performed a Y2H assay to test the interaction of LIN-2's PDZ and SH3 domains with ACR-16. Our results showed that the SH3 but not the PDZ domain directly binds to a cytoplasmic loop of ACR-16 (Fig 7A and 7B). Interestingly, the SH3 domain also mediates the interaction between LIN-2 and UNC-29 (Fig 7A and 7D). To test the role of other LIN-2 domains (e.g., CaM kinase, L27, and GK; Fig 1), we expressed a truncated LIN-2 protein lacking the SH3 domain (LIN-2ΔSH3), and found that LIN-2ΔSH3 did not detectably interact with ACR-16 and UNC-29 (Fig 7A and 7B). In contrast, LIN-2 lacking the PDZ domain (LIN-2ΔPDZ) still exhibited binding activity to both receptors (Fig 7A and 7B), demonstrating that the SH3 is the only domain mediating interaction with ACR-16 and UNC-29. Together, our results reveal that the LIN-2/FRM-3 complexes regulate the synaptic abundance of ionotropic AChRs, likely through binding and immobilizing clusters of AChRs on the postsynaptic cell surface.

## The PDZ domain in LIN-2/CASK and the FERM domain in FRM-3/FARP are required for LIN-2/FRM-3 interaction

To understand how the LIN-2/FRM-3 complex is formed, we examined what domains in LIN-2 and FRM-3 mediate their interaction. In LIN-2, we focused on the central PDZ and SH3 domains. Consistent with our previous findings, strong interaction was observed between full-length LIN-2 and FRM-3 (Fig 7A and 7F) [20]. No interaction was detected between the SH3 domain and FRM-3, indicating that SH3 does not mediate the LIN-2/FRM-3 interaction (Fig 7A and 7F). This was confirmed by the LIN-2ΔSH3, which still exhibited strong binding to FRM-3 (Fig 7A and 7F). Like the SH3 domain, the isolated PDZ domain also did not exhibit binding to FRM-3. However, the LIN-2ΔPDZ failed to bind to FRM-3 (Fig 7A and 7F). This

## A

### Yeast two-hybrid assay

| | BDFRM-3 | BDFERM | BDLIN-2A | BDLIN-2A-SH3 | BDLIN-2AΔSH3 | BDLIN-2A-PDZ | BDLIN-2AΔPDZ |
|---|---|---|---|---|---|---|---|
| ADACR-16 LoopII | - | - | ++ | ++ | - | - | ++ |
| ADUNC-29 LoopII | - | - | + | + | - | - | + |
| ADFRM-3 | | | ++ | - | ++ | - | - |
| ADFERM | | | ++ | - | ++ | - | - |

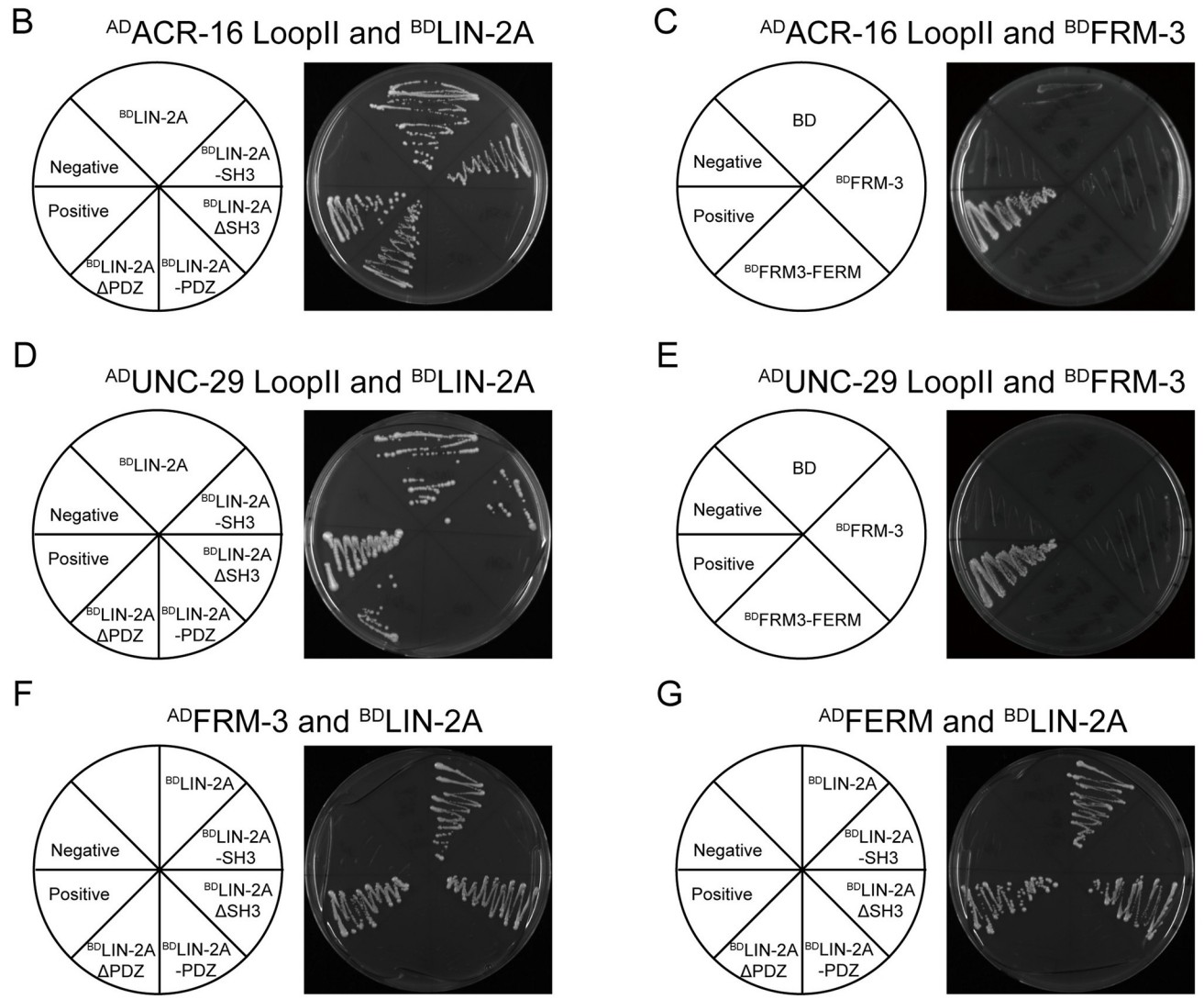

**Fig 7. LIN-2/CASK binds to both ACR-16 and UNC-29 through SH3 domain.** (A) Summary of interactions by Yeast two-hybrid. Strong interaction (++); weak interaction (+), and no interactions (-) were indicated. (B) LIN-2A's SH3 domain binds the ACR-16's second intracellular loop (LoopII) in a Yeast two-hybrid assay. Y2HGold cells carrying indicated plasmids (Left) growing on selective media (-Trp/-Leu/-His/-Ade) is shown (Right). (C) LIN-2A's SH3 domain binds the UNC-29's second intracellular loop (LoopII) in the Yeast two-hybrid assay. (D-E) FRM-3 do not bind the ACR-16's second intracellular loop (LoopII) (D) and UNC-29's second intracellular loop (LoopII) (E) in the Yeast two-hybrid assay. (F-G) LIN-2A binds FRM-3 (F) and its FERM domain (G) requiring its PDZ domain, but not SH3 domain.

indicates that the PDZ domain is required for LIN-2/FRM-3 interaction, although it alone is not sufficient to mediate this interaction. In FRM-3, the FERM domain exhibited strong binding to both the full-length LIN-2 and the LIN-2ΔSH3, but not the SH3, PDZ, and LIN-2ΔPDZ (Fig 7A and 7G), similar to the binding pattern seen with full-length FRM-3. Taken together, our results demonstrate that LIN-2 binds to the FERM domain in FRM-3, and this interaction requires the PDZ domain in LIN-2.

## Cholinergic transmission is decreased by deleting the SH3 or the PDZ domain in LIN-2/CASK

The interaction between LIN-2 SH3 and AChRs prompted us to investigate the functional importance of this domain. We deleted SH3 by CRISPR and analyzed synaptic transmission (Fig 8A; *syb1921*, ΔSH3). Our data showed that cholinergic transmission was dramatically decreased in ΔSH3 mutants, including mEPSC frequency and amplitude, evoked EPSC amplitude and charge transfer, and mEPSC charge (Fig 8B–8F). These results indicate that the SH3 domain is indispensable for LIN-2's function, and the cholinergic synaptic transmission is determined by its interaction with AChRs. As the PDZ domain is required for LIN-2/FRM-3 interaction, we also tested whether it participates in the regulation of cholinergic synaptic transmission. Interestingly, deleting the PDZ domain (*syb1937*, ΔPDZ; Fig 8A) caused similarly stronger decrease in cholinergic transmission, demonstrating that the PDZ domain also plays essential roles in LIN-2's synaptic function. Thus, our data suggest that the function of LIN-2/CASK in cholinergic synaptic transmission requires the PDZ domain, which recruits FRM-3/FARP, allowing LIN-2/FRM-3 complexes to bind AChRs via LIN-2's SH3 domain.

The functional importance of the SH3 and PDZ domains was also evaluated in GABAergic synapses. The mIPSC amplitude in ΔSH3 and ΔPDZ mutants was decreased to a similar extent to that in *lin-2* null mutants (Fig 8F and 8G). However, the mIPSC frequency was not changed in the ΔSH3 and ΔPDZ mutants, unlike the *lin-2* null mutants (Fig 2). These results suggest that the mutant LIN-2 proteins are expressed and retain some function.

Collectively, our findings provide novel and interesting models regarding the functions of scaffolding proteins LIN-2/CASK and FRM-3/FARP in *C. elegans* NMJ (Fig 9). We demonstrate that LIN-2 and FRM-3 regulate the clustering and/or stabilization of nAChRs and L-AChRs in body wall muscles, without affecting their overall expression or delivery to the cell surface. These functions require the participation of the SH3 and PDZ domains of LIN-2, which mediate interaction with AChRs and FRM-3. At the GABAergic synapses, postsynaptic LIN-2 and FRM-3 are also involved in the regulation of GABA$_A$Rs. Moreover, the delivery of GABA$_A$Rs to the cell surface requires LIN-2 but not FRM-3.

## Discussion

Different from the NMJs in mouse and fly which use ACh or glutamate as their primary neurotransmitters, the *C. elegans* NMJs comprise both cholinergic and GABAergic synapses [39]. This allows us to use the worm NMJ to investigate three different neurotransmitter receptors, nAChRs, L-AChRs, and GABA$_A$Rs. Moreover, the major known synaptic organizers found in vertebrates, such as PSD-95, rapsyn, MuSK, and neuroligin, also exist in worm NMJ, suggesting that the regulatory mechanisms of the AChRs and GABA$_A$Rs in *C. elegans* are similar to those in other species. In this study, we investigated the mechanisms by which LIN-2/CASK and FRM-3/FARP regulate cholinergic synaptic transmission at the *C. elegans* NMJ. Our results lead to several previously unknown findings. First, the *lin-2* and *frm-3* null mutants

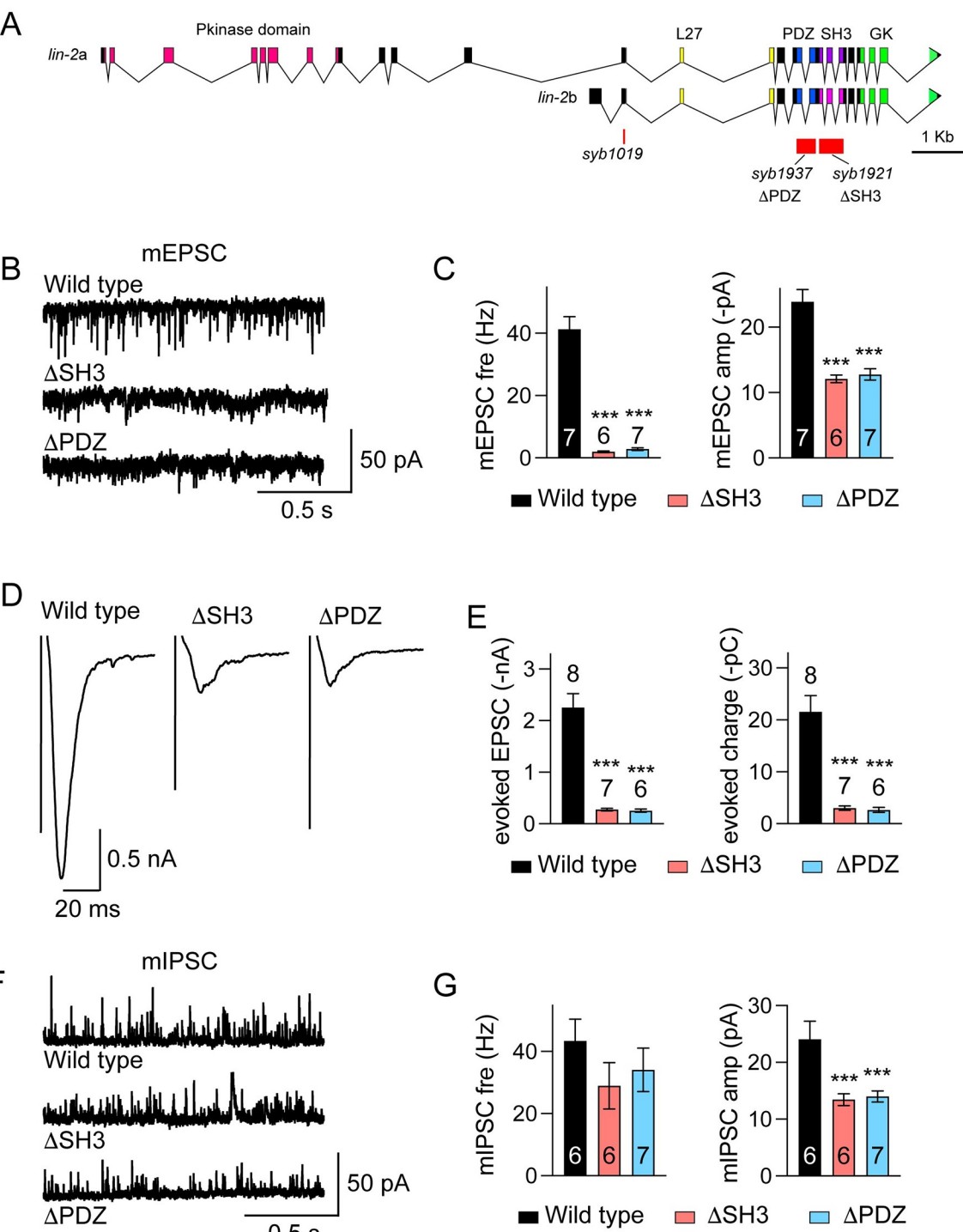

**Fig 8. The SH3 and PDZ domains are indispensable for LIN-2 function.** (A) The SH3 domain and the PDZ domain are deleted in the endogenous *lin-2* gene by crispr. (B-G) Cholinergic and GABAergic synaptic transmission were severely reduced in *lin-2* ΔSH3 and ΔPDZ mutants. Representative traces of mEPSCs (B), evoked EPSCs (D), mIPSCs (F), quantitation of mEPSC frequency and amplitude (C), the amplitude and charge transfer of evoked EPSCs (E), and mIPSC frequency and amplitude (G) are shown. Data are mean ± SEM (***, $p < 0.001$ when compared to wild type; one-way ANOVA). The number of worms analyzed for each genotype is indicated in the bar.

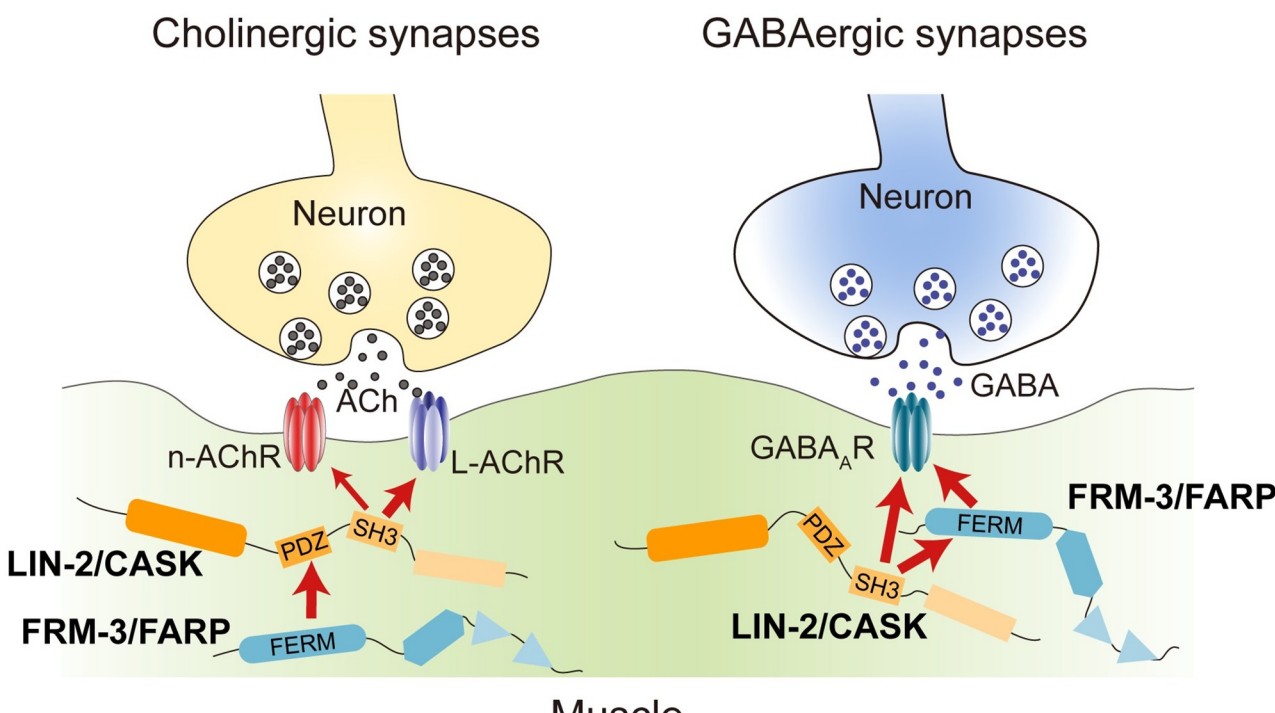

**Fig 9. Model for LIN-2/CASK and FRM-3/FARP complex in cholinergic and GABAergic synapses.** See discussion for details.

exhibit severe defects in cholinergic synaptic transmission and locomotion behavior. Second, the cholinergic transmission defects are caused by the loss of function of LIN-2 and FRM-3 in postsynaptic body wall muscle but not presynaptic neurons. Third, LIN-2 and FRM-3 regulate the synaptic abundance of two classes of ionotropic AChRs (nAChRs and L-AChRs) on the muscle surface. Fourth, SH3 domain in LIN-2 physically binds to both classes of ionotropic AChRs, and deleting this domain causes strong transmission defects. These findings reveal interesting and important regulatory mechanisms for synaptic transmission at cholinergic synapses. Below we discuss the significance of these findings.

## CASK and FARP regulate cholinergic synaptic transmission

Since its discovery as a neurexin-binding protein [29], CASK has been found to be broadly involved in the regulation of cell-cell junction organization. The high homology of the C-terminal domains in CASK (e.g., PDZ, SH3, and GK) with other MAGUK proteins such as PSD95 has suggested that CASK may function similarly by acting as a scaffolding protein, while the presence of the unique N-terminal CaM kinase domain endows CASK additional functions in various cellular pathways [45]. In addition to neurexin, CASK also binds to many other membrane proteins, such as Mint-1, Velis, Syndecan, FARP, glycophorins, and SynCaM through its central PDZ domain [46–49]. The CASK homolog in *C. elegans* LIN-2 is required for vulva development and proper localization of the EGF receptor LET-23 [34,50].

Despite its functions in membrane organization, the physiological importance of CASK in synapses has been controversial. Knockout of CASK in mice causes lethal phenotype but does not affect the development of neurons. Cortical neurons lacking CASK display normal

ultrastructural morphology and evoked EPSCs, whereas spontaneous glutamate release is increased but spontaneous GABA release is decreased [33]. In *Drosophila*, loss of CAKI, the CASK ortholog, leads to increased frequency of MEPPs without changes in MEPP amplitude [51]. However, a later study in *Drosophila* found reduced spontaneous and evoked neurotransmitter release, as well as decreased postsynaptic glutamate receptors in CASK knockout mutants [18].

FARP was first recognized as a synaptic component that is required for dendrite growth in spinal motor neurons [52]. Later studies revealed that FARP forms a synaptic complex with SynCAM1 through its FERM domain and functions as a synaptic organizer in cultured dissociated hippocampal neurons [53]. It is enriched at postsynaptic sites and increases synapse number and modulates spine morphology. At *C. elegans* NMJs, FRM-3/FARP and its binding protein LIN-2/CASK promote post-synaptic localization of $GABA_AR$, most likely by immobilizing receptors at post-synaptic specializations [20,21]. Here we describe LIN-2/CASK and FRM-3/FARP function at cholinergic synapses, finding that they regulate postsynaptic abundance of ionotropic AChRs.

Several lines of evidence suggest that LIN-2/CASK and FRM-3/FARP form a scaffold complex regulating synaptic transmission at the *C. elegans* NMJ. First, these two scaffolds physically bind each other [20,21]. Second, synaptic transmission defects in *lin-2* and *frm-3* mutants are comparable, and no additional decrease in the *lin-2; frm-3* double mutants (Fig 1), demonstrating that they act in the same genetic pathway. Third, both scaffolds function in postsynaptic body wall muscle regulating the synaptic abundance of two classes of AChRs (Fig 4) [24]. Although post-synaptic currents are dramatically reduced, locomotion rates were only modestly reduced in *lin-2* and *frm-3* null mutants. This persistent locomotion behavior likely reflects the residual function of post-synaptic L-AChRs in these mutants. Prior studies of the *Drosophila* NMJ have proposed that neurexin may also be involved in the CASK and protein 4.1 complex (CASK/neurexin/4.1), which regulates glutamate receptor clustering [18]. However, it seems that the LIN-2/FRM-3 complex does not require neurexin for clustering and stabilizing ionotropic AChRs at the worm NMJ, as the worm mutants lacking *nrx-1*/neurexin exhibit normal mEPSCs and evoked EPSCs [20,54]. A recent study showed that the loss of γ-neurexin causes a moderate decrease in evoked EPSCs at the NMJ; however, the mEPSCs, including frequency and amplitude, are unchanged, indicating that the ionotropic AChRs are functionally normal [55].

Despite their expression in motor neurons, our data show that LIN-2 and FRM-3 are not required for SV fusion at the presynaptic terminals, evidenced by the normal mEPSCs and evoked EPSCs in *lin-2* and *frm-3* neuron cKO mutants (Fig 3). It remains possible that LIN-2 and FRM-3 may play important roles in other neuronal functions, e.g. promoting the localization of presynaptic inotropic receptors.

## Functional roles of LIN-2A and LIN-2B isoforms in synaptic transmission

The *C. elegans lin-2* locus expresses two isoforms: a long isoform (LIN-2A) that includes the N-terminal CaM kinase domain and a short isoform (LIN-2B) lacking this domain (Fig 1) [34]. It is believed that the *lin-2 e1309* mutants express only the LIN-2B isoform, as the deletion removes the whole promoter region for LIN-2A and part of the CaM kinase domain (Fig 1A). Our data show that LIN-2A plays a more dominant role in cholinergic transmission, accounting for 70% of the evoked neurotransmitter release. The more severe decrease in cholinergic synaptic transmission including both mEPSCs and evoked EPSCs in the *lin-2* null mutants demonstrates that the LIN-2B isoform also plays important roles in synaptic

function. In GABAergic synapses, it seems the LIN-2B isoform plays similarly important role as LIN-2A, as the mIPSC amplitude was decreased by 20% in the *e1309* mutants and was decreased by a further 20% in the null mutants, different from the observations in mEPSC amplitude (30% decrease in *e1309* and 40% decrease in null). Moreover, the mIPSC frequency was severely decreased in the null mutants but was not changed in the *e1309* mutants. These results suggest that LIN-2A and LIN-2B may function redundantly in GABAergic synapses (Fig 2D). A *lin-2b* specific knockout mutant may help to confirm this notion in future studies.

Our data suggest that both LIN-2A and LIN-2B are involved in the regulation of AChRs clustering and stabilization in cholinergic synapses, consistent with results reported in a recent study [24]. Knockout of either LIN-2A (*e1309* mutant) or both LIN-2A and LIN-2B causes decreased mEPSC amplitude without altering puffing ACh- or Levamisole-activated currents (Figs 1 and 4), indicating normal AChRs trafficking and surface delivery. However, LIN-2A and LIN-2B appear to function differently in regulating $GABA_A$Rs. The decreased mIPSC amplitude but normal puffing GABA-activated currents in the *e1309* mutants [20], demonstrate that LIN-2A functions only for $GABA_A$Rs clustering and stabilization. In contrast, the *lin-2* null mutants exhibited a 50% reduction of puffing GABA-activated currents (S2 Fig), indicating that the LIN-2B isoform is more likely involved in the delivery or retention of $GABA_A$Rs on the muscle surface. These results suggest that the two LIN-2 isoforms have different functions at cholinergic and GABAergic synapses. Our data do not distinguish the two FRM-3 isoforms, as both are disrupted in *gk585* and null mutants. The impact of LIN-2 and FRM-3 on post-synaptic receptor clustering could be mediated (at least in part) by syndecan. This is supported by the recent findings that *sdn-1*/syndecan regulates the synaptic clustering of AChRs by recruiting LIN-2/CASK and FRM-3/FARP at the *C. elegans* cholinergic NMJs [24]. Future studies will investigate these possibilities, as well as other potential LIN-2/FRM-3 binding partners in organizing post-synaptic receptor clustering.

Several results suggest that the deficits in synaptic transmission observed in *lin-2* and *frm-3* mutants arise primarily from failure to cluster post-synaptic ACh and GABA receptors. Muscle specific *lin-2* and *frm-3* knockouts recapitulate the ACh and GABA NMJ electrophysiological and post-synaptic receptor clustering defects observed in the corresponding null mutants, whereas neuron specific knockouts had no effects. The ultrastructure of ACh and GABA pre-synaptic terminals and the localization of several pre-synaptic markers was unaffected in *lin-2* and *frm-3* null mutants. Collectively, these results suggest that pre-synaptic structure and function are largely unaffected by inactivating LIN-2 and FRM-3. Our results do not exclude the possibility that further analysis will reveal pre-synaptic functions for these scaffolding proteins, as has been described for mammalian synapses [33].

## Materials and methods

### Strains

Animals were cultivated at room temperature on nematode growth medium (NGM) agar plates seeded with OP50 bacteria. On the day before experiments L4 larval stage animals were transferred to fresh plates seeded with OP50 bacteria for all the electrophysiological, imaging, and behavioural experiments. The following strains were used:

Wild type, N2 bristol

CB1309 *lin-2(e1309)*

PHX1019 *lin-2(syb1019)*

KP7338 *frm-3(gk585)*

PHX1036 *frm-3(syb1036)*

ZTH1032 *lin-2(e1309);frm-3(gk585)*

MT106 *lin-7(n106)*

KP7637 *lin-10(n1508)*

KP10507 *lin-2(nu743, flex ON)*

KP10609 *lin-2(nu743, flex ON)*; *nuSi502[Psbt-1::NLS-CRE::SL2::NLS-BFP]*

KP10621 *lin-2(nu743, flex ON)*; *nuSi491[Pmyo-3::CRE]*

KP10590 *frm-3(nu751 FLEX ON)*

KP10676 *frm-3(nu751 FLEX ON)*; *nuSi502[Psbt-1::NLS-CRE::SL2::NLS-BFP]*

KP10677 *frm-3(nu751 FLEX ON)*; *nuSi491[Pmyo-3::CRE]*

KP9579 *unc-2(nu657, flex ON)*

KP10796 *unc-2(nu657)*; *nuSi502[Psbt-1::NLS-CRE::SL2::NLS-BFP]*

TXJ0502 *nuSi0002[Pmyo-3::ACR-16::RFP]*

TXJ0825 *lin-2(syb1019)*; *nuSi0002[Pmyo-3::ACR-16::RFP]*

TXJ0822 *frm-3(gk585)*; *nuSi0002[Pmyo-3::ACR-16::RFP]*

EN208 *kr208 [Punc-29::UNC-29::tagRFP]*

TXJ0569 *lin-2(syb1019)*; *kr208[Punc-29::UNC-29::tagRFP]*

TXJ0816 *frm-3(gk585)*; *kr208[Punc-29::UNC-29::tagRFP]*

KP9809 *nu586[UNC-2::GFP11x7]*; *nuSi250[Punc-129::splitGFP1-10::SL2::UNC-57-mCherry::SL2::mTagBFP2]*

TXJ0818 *lin-2(syb1019)*; *nu586[UNC-2::GFP11x7]*; *nuSi250[Punc-129::splitGFP1-10::SL2::UNC-57-mCherry::SL2::mTagBFP2]*

TXJ1283 *frm-3(syb1036)*; *nu586[UNC-2::GFP11x7]*; *nuSi250[Punc-129::splitGFP1-10::SL2::UNC-57-mCherry::SL2::mTagBFP2]*

PHX1921 *lin-2 ΔSH3 (syb1921)*

PHX1937 *lin-2 ΔPDZ (syb1937)*

## Generation of *lin-2* and *frm-3* mutants

*lin-2(syb1019)* contains an opal stop codon at the first common region of LIN-2A (E420) or LIN-2B (E79). Forward genotype primer: tagtatccagcccgacgagt, reverse primer: gatgtgtggc-taacgggtga. *frm-3(syb1036)* contains an amber stop codon at the first common region of FRM-3A (V36) or FRM-3B (V48). Forward genotype rimer: ccaaccaggtgcggatcata, reverse primer: cgagaagttgctcacctggt. *lin-2 ΔSH3 (syb1921)* contains a deletion from F637 to A718 for LIN-2A, or F296 to A377 for LIN-2B. *lin-2 ΔPDZ (syb1937)* contains a deletion from R545 to T620 for LIN-2A, or R204 to T279 for LIN-2B.

## Transgenes and germline transformation

Transgenic strains were isolated by microinjection of various plasmids using either Pmyo-2::NLS-GFP (KP#1106) or Pmyo-2::NLS-mCherry (KP#1480) as the co-injection marker. The single copy insertion lines were generated using miniMos transposon [56].

## CRISPR alleles

CRISPR alleles were isolated as described [57]. Briefly, *unc-58* was used as a co-CRISPR selection to identify edited animals. Animals were injected with two guide RNAs (gRNAs) and two repair templates, one introducing an *unc-58* gain of function mutation and a second modifying a gene of interest. Progeny exhibiting the *unc-58(gf)* uncoordinated phenotype were screened for successful editing of the second locus by PCR. Split GFP and split sfCherry constructs are described in [58].

MiniMOS inserts in which Pmyo-3 drives expression of either GFP 1–10 (nuTi144) or sfCherry 1–10 SL2 GFP 1–10 (nuTi458) were created using the protocol as described by [56].

Tissue specific *frm-3* and *lin-2* knockout was performed by introducing a stop cassette into introns in the ON configuration (i.e. in the opposite orientation of the target gene) using CRISPR, creating the *frm-3(nu751, flex ON)* and *lin-2(nu743, flex ON)* alleles. In *frm-3(nu751, flex ON)*, the stop cassette was inserted into an intron shared by *frm-3a* and *b* (intron 3 of *frm-3a*). Similarly, the stop cassette in *lin-2(nu743, flex ON)* was inserted into an intron shared by both *lin-2* isoforms (intron 3 of *lin-2b*). The stop cassette consists of a synthetic exon (containing a consensus splice acceptor sequence and stop codons in all reading frames) followed by a 3' UTR and transcriptional terminator taken from the *flp-28* gene (the 564bp sequence just 3' to the *flp-28* stop codon). The stop cassette is flanked by FLEX sites (which are modified loxP sites that mediate CRE induced inversions) [59]. In this manner, orientation of the stop cassette within the intron is controlled by CRE expression. Expression of the targeted gene is reduced when the stop cassette is in the OFF configuration (i.e. the same orientation as the targeted gene) but is unaffected in the ON configuration (opposite orientation). The endogenous *flp-28* gene is located in an intron of W07E11.1 (in the opposite orientation). Consequently, we reasoned that the *flp-28* transcriptional terminator would interfere with *frm-3* and *lin-2* expression in an orientation selective manner. A similar strategy was previously described for conditional gene knockouts in *Drosophila* [60].

## Locomotion and behavioral assays

Young adult animals were washed with a drop of PBS and then transferred to fresh NGM plates with no bacterial lawn (30 worms per plate). Worm movement recordings (under room temperature 22˚C) were started 10 min after the worms were transferred. A 2 min digital video of each plate was captured at 3.75 Hz frame rate by WormLab System (MBF Bioscience). Average speed and tracks were generated for each animal using WormLab software.

## Fluorescence imaging

Worms were immobilized by 30 g/l 2,3-Butanedione monoxime (Sigma) and mounted on 2% agar on glass slides. Fluorescence images were captured by 100x (NA = 1.4) objective on an Olympus microscope (BX53). The mean fluorescence intensities of reference FluoSphere microspheres (Thermo Fisher Scientific) were measured during each experiment, and were used to control for illumination intensities changes. Multidimensional data were reconstructed as maximum intensity projections using Metamorph software (Molecular Devices). Line scans

were analyzed in Igor Pro (WaveMetrics) using a custom script. The intensity of peak fluorescence of each marker was analyzed.

## Electrophysiology

Electrophysiology was conducted on dissected *C. elegans* as previously described [54]. Worms were superfused in an extracellular solution containing 127 mM NaCl, 5 mM KCl, 26 mM NaHCO$_3$, 1.25 mM NaH$_2$PO$_4$, 20 mM glucose, 1 mM CaCl$_2$, and 4 mM MgCl$_2$, bubbled with 5% CO$_2$, 95% O$_2$ at 22˚C. Whole-cell recordings were carried out at -60 mV for all EPSCs, including mEPSCs and evoked EPSCs. The holding potential was switched to 0mV to record mIPSCs. The internal solution contained 105 mM CH$_3$O$_3$SCs, 10 mM CsCl, 15 mM CsF, 4mM MgCl$_2$, 5mM EGTA, 0.25mM CaCl$_2$, 10mM HEPES, and 4mM Na$_2$ATP, adjusted to pH 7.2 using CsOH. Stimulus-evoked EPSCs were obtained by placing a borosilicate pipette (5–10 μm) near the ventral nerve cord (one muscle distance from the recording pipette) and applying a 0.4 ms, 85μA square pulse. Sucrose-evoked release was triggered by a 2s application of 0.5M sucrose dissolved in normal bath solution. For drug-activated current recordings, ACh, Levamisole, or GABA was pressure injected onto body wall muscle. All recordings were performed at room temperature (22˚C).

## Electron microscopy

Samples were prepared using high-pressure freeze fixation [61]. ~30 young adult hermaphrodites were placed in each specimen chamber containing *E. coli* and were frozen at -180˚C under high pressure (Leica SPF HPM 100). Frozen specimens then underwent freeze substitution (Leica Reichert AFS) during which the samples were held at −90˚C for 107 h in 0.1% tannic acid and 2% OsO$_4$ in anhydrous acetone. The temperature was then increased from 5˚C/h to −20˚C, kept at −20˚C for 14 h, and increased by 10˚C/h to 20˚C. After fixation, samples were infiltrated with 50% Epon/acetone for 4 h, 90% Epon/acetone for 18 h, and 100% Epon for 5 hr. Finally, samples were embedded in Epon and incubated for 48 h at 65˚C. Ultra-thin serial sections (50 nm) were cut and glued to a wafer, and counterstained in 0.08 mol/L citrate for 10 minutes. Images were acquired using a GeminiSEM460 scanning electron microscope operating at 5 kV. Images were collected from the ventral and dorsal nerve cord region anterior to the vulva for all genotypes. Cholinergic synapses were identified on the basis of their typical morphology and anatomical features [35]. A synapse was defined as a series of sections (profiles) containing a dense projection as well as two flanking sections on both sides without dense projections. Image acquisition and analysis using NIH ImageJ/Fiji software were performed blinded for genotype.

## Yeast two-hybrid

The yeast two-hybrid assay was performed with the Matchmaker Gold Yeast Two-Hybrid System. Briefly, LIN-2A, LIN-2A-SH3, LIN-2A-PDZ, LIN-2A-ΔSH3, LIN-2A-ΔPDZ, FRM-3 and FERM were cloned into pGBKT7 as a bait vector. In addition, ACR-16-TM3-4(aa315-472) and UNC-29-TM3-4(aa318-445) were cloned into pGADT7 as a prey vector. The plasmid pairs were simultaneously transformed into the Y2HGold yeast cells. Transformants selected from the SD-Leu-Trp plates were restreaked onto SD-Leu-Trp-His-Ade plates to test interactions. Meanwhile, the pGBKT7-53 and pGADT7-T plasmids pair were used as positive controls, and pGBKT7-Lam and pGADT7-T plasmids pair were used as negative controls. False-positive from autoactivation was ruled out by co-transformation of pGBKT7-LIN-2A construct with pGADT7 empty vector alone.

## Plasmids

| KP2561 | pGADT7-ACR-16-TM3-4(aa315-472) |
|--------|-------------------------------|
| PXJ461 | pGADT7-UNC-29-TM3-4(aa318-445) |
| KP1890 | pGBKT7-LIN-2A |
| PXJ278 | pGBKT7-LIN-2A-SH3 |
| PXJ279 | pGBKT7-LIN-2A-PDZ |
| PXJ463 | pGBKT7-LIN-2A-ΔSH3 |
| PXJ464 | pGBKT7-LIN-2A-ΔPDZ |
| PXJ413 | pGBKT7-FRM-3 |
| KP1950 | pGBKT7-FERM |

## Data acquisition and statistical analysis

All electrophysiological data were obtained using a HEKA EPC10 double amplifier (HEKA Elektronik) filtered at 2 kHz, and analyzed with open-source scripts developed by Eugene Mosharov (http://sulzerlab.org/Quanta_Analysis_8_20.ipf) in Igor Pro 7 (Wavemetrics). All imaging data were analyzed in ImageJ software. Each set of data represents the mean ± SEM of an indicated number (n) of animals. To analyze mEPSCs and mIPSCs, a 3.5pA peak threshold was preset, above which release events are clearly distinguished from background noise. The analyzed results were re-checked by eye to ensure that the release events were accurately selected.

## Statistical analysis

All data were statistically analyzed in Prism 8 software. Normality distribution of the data was determined by the D'Agostino-Pearson normality test. When the data followed a normal distribution, an unpaired student's t-test (two-tailed) or one-way ANOVA was used to evaluate the statistical significance. In other cases, a Mann-Whitney test or one-way ANOVA following Kruskal-Wallis test was used.

## Supporting information

**S1 Fig. Cholinergic synaptic transmission is normal in *lin-7* and *lin-10* mutants.** mEPSCs and evoked EPSCs were recorded in *lin-7* and *lin-10* mutants. Representative traces of mEPSCs and evoked EPSCs (A, C), and mean mEPSC frequency and amplitude, and evoked EPSC amplitude and charge transfer are shown. Data are mean ± SEM. The number of worms analyzed for each genotype is indicated in the bar.
(TIF)

**S2 Fig. mEPSC frequency is decreased in *acr-16* mutants.** (A, B) Quantitation of mEPSC frequency and amplitude in wild type and *acr-16* mutants. (C) Despite the severe defects in mEPSCs, *acr-16* mutants exhibit normal locomotion speed. Data are mean ± SEM (***, $p < 0.001$ when compared to wild type; student's t-test). The number of worms analyzed for each genotype is indicated in the bar.
(TIF)

**S3 Fig. Locomotion speed in *lin-2* and *frm-3* conditional knockout mutants.** Locomotion was analyzed in two independent experiments in two days. The speed was decreased in *lin-2* Muscle^Cre mutants but unchanged in *frm-3* Neuron^Cre and Muscle^Cre mutants. Data are

mean ± SEM (\*\*\*, $p < 0.001$ when compared to Control animals; n.s., non-significant; one-way ANOVA). The number of worms analyzed for each genotype is indicated in the bar.
(TIF)

**S4 Fig. The CRE recombinase inactivates *unc-2* in neurons.** (A) Averaged speed in *unc-2*<sup>FLEX</sup>(*nu657*) mutants, wild type animals expressed Neuron<sup>Cre</sup>, and *unc-2*<sup>FLEX</sup>(*nu657*) expressed Neuron<sup>Cre</sup>. (B, C) Example traces of mEPSCs and summary of mEPSC frequency and amplitude from the same genotypes in A. (D, E) mIPSC traces and quantification of mIPSC frequency and amplitude. (F, G) Evoked EPSC traces and quantification of EPSC amplitude and charge transfer. Data are mean ± SEM (\*\*\*, $p < 0.001$ when compared to *unc-2*<sup>FLEX</sup>(*nu657*); one-way ANOVA). The number of worms analyzed for each genotype is indicated in the bar.
(TIF)

**S5 Fig. Loss of function of *lin-2* and *frm-3* affect the decay of mEPSCs.** (A, B) Example of averaged mEPSCs and mIPSCs. Each current was averaged by all mini events in a single mEPSC or mIPSC trace (15 sec). (C) Quantification of the decay of the averaged mEPSCs and mIPSCs. Data are mean ± SEM (\*\*\*, $p < 0.001$ when compared to wild type; n.s., non-significant; one-way ANOVA). The number of worms analyzed for each genotype is indicated under each box.
(TIF)

**S6 Fig. Surface level of AChRs and GABA$_A$Rs in *lin-2* and *frm-3* cKO mutants.** (A, C, E) Representative traces of ACh-, GABA-, and levamisole-activated currents in neuron and muscle conditional knockout mutants of *lin-2* and *frm-3*. (B, D, F) Quantification of current amplitude. Data are mean ± SEM (\*\*\*, $p < 0.001$ when compared to control; one-way ANOVA). The number of worms analyzed for each genotype is indicated in the bar.
(TIF)

**S1 Table. Summary of the locomotion speed data in this study.** Data are presented as mean ± SEM.
(PDF)

**S2 Table. Summary of the spontaneous and evoked EPSC data in this study.** Data are presented as mean ± SEM.
(PDF)

**S3 Table. Summary of the imaging data in this study.** Data are presented as mean ± SEM.
(PDF)

**S4 Table. Summary of ACh, Levamisole, and GABA-activated currents in this study.** Data are presented as mean ± SEM.
(PDF)

**S5 Table. Summary of mini decay.** Data are presented as mean ± SEM.
(PDF)

## Acknowledgments

We thank the *C. elegans* Genetics Stock Center for providing strains. We also thank the Molecular Imaging Core Facility (MICF) at the School of Life Science and Technology, Shanghai-Tech University for help in imaging.

## Author Contributions

**Conceptualization:** Lei Li, Haowen Liu, Kang-Ying Qian, Stephen Nurrish.

**Formal analysis:** Lei Li, Haowen Liu, Kang-Ying Qian, Stephen Nurrish.

**Funding acquisition:** Joshua M. Kaplan, Xia-Jing Tong, Zhitao Hu.

**Investigation:** Lei Li, Haowen Liu, Kang-Ying Qian, Stephen Nurrish, Xian-Ting Zeng, Wan-Xin Zeng, Jiafan Wang, Joshua M. Kaplan, Xia-Jing Tong, Zhitao Hu.

**Project administration:** Joshua M. Kaplan, Xia-Jing Tong, Zhitao Hu.

**Supervision:** Joshua M. Kaplan, Xia-Jing Tong, Zhitao Hu.

**Writing – original draft:** Joshua M. Kaplan, Xia-Jing Tong, Zhitao Hu.

**Writing – review & editing:** Lei Li, Haowen Liu, Kang-Ying Qian, Stephen Nurrish, Joshua M. Kaplan, Xia-Jing Tong, Zhitao Hu.

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
