## [Decision Letter · Decision Letter 0]

19 May 2022

Dear Dr Hu,

Thank you very much for submitting your Research Article entitled 'CASK and FARP coordinately control pre- and post-synaptic function' to PLOS Genetics.

The manuscript was fully evaluated at the editorial level and by independent peer reviewers. The reviewers appreciated the attention to an important problem, but raised some substantial concerns about the current manuscript. Based on the reviews, we will not be able to accept this version of the manuscript, but we would be willing to review a much-revised version. We cannot, of course, promise publication at that time.

If you decide to revise the manuscript for further consideration at PLOS Genetics, please aim to resubmit within the next 90 days, unless it will take extra time to address the concerns of the reviewers, in which case we would appreciate an expected resubmission date by email to plosgenetics@plos.org.

[LINK]

Please do not hesitate to contact us if you have any concerns or questions.

Yours sincerely,

Dion Kai Dickman, PhD

Guest Editor

PLOS Genetics

Gregory P. Copenhaver

Editor-in-Chief

PLOS Genetics

Dear Dr. Hu,

Thank you for submitting your manuscript to PLoS Genetics. As you will see below, your manuscript was reviewed by three reviewers. Overall the reviews were positive and noted the powerful combination of synergistic approaches and genetic analyses. That being said and as detailed below, some significant criticisms were raised. Please respond carefully to the issues raised by the reviewers, particularly the genetic controls and questions about GABA vs ACh raised by reviewers 1 and 3, in preparing a revised manuscript if you so choose. We are looking forward to receiving your revised manuscript.

Reviewer's Responses to Questions

**Comments to the Authors:**

Reviewer #1: please see attached review

Reviewer #2: This paper reports how C. elegans scaffolding proteins LIN-2 (a CASK homolog) and FRM-3 (a FARP homolog) regulate the function of NMJ synapses. C. elegans NMJ synapses are cholinergic. The authors make an argument in the Abstract of the paper that little is known about how scaffolds affect the development and function of cholinergic synapses. This is in contrast to information known about other excitatory and inhibitory synapses like glutamatergic or GABAergic synapses. By this framing, a characterization of LIN-2 and FRM-3 should expand our knowledge about cholinergic synapses and neurotransmission.

The first findings the authors report are behavioral and electrophysiological. Loss of either the lin-2 or frm-3 genes impairs locomotion and NMJ transmission. Double mutant combinations seem no more severe than single mutant combinations, suggesting a shared process or pathway for LIN-2 and FRM-3. Collectively, the authors report:

• behavior/locomotion data

• electrophysiological characterizations

• new null alleles that they generated for this study

• airtight genetic data localizing the electrophysiological functions to the muscle

• clear effects on acetylcholine receptors by imaging (suggesting impaired clustering)

• EM and fluorescence microscopy, suggesting that synapse development is normal

• meaningful distinctions between the cholinergic synapse roles and the GABAergic roles of these proteins

• structure and function and binding experiments to extend the above, suggesting importance for the PDZ domain of LIN-2 and the FERM domain of FRM-3 for binding – and specific domains of LIN-2 for function (PDZ and SH3)

Taking the data and results together, the authors have assembled an impressive and convincing dataset demonstrating that postsynaptic LIN-2 and FRM-3 are important for NMJ function. In this reviewer’s opinion, this paper should interest many readers of PLOS Genetics, including synaptic physiologists and neuroscientists interested in synapse development. The authors generated a lot of valuable reagents for the field, especially new null alleles which shed light upon the functions of LIN-2 and FRM-3.

There are a few points that could be cleared up with text revisions.

Main Points

1. In terms of framing, there is a disconnect on what is known from prior work. The Abstract states, “far less is known about how scaffolds shape cholinergic synapses.” Similarly, an early point of the Introduction reads, “much less is known about how nicotinic receptors are localized to synapses.” But then immediately after that, the authors correctly list a whole panoply of factors known to stabilize AChR synapses (sentences that span pp. 3-4). This reads in an incongruous way.

LIN-2 and FRM-3 add to the body of knowledge, and their roles at the C. elegans NMJ are worth understanding. But as the text is currently written in the Introduction, it is unclear what major gaps in understanding are missing at cholinergic synapses in contrast to other types of synapses. It might be worth defining those gaps a little more clearly in the Introduction and then revisiting them in a Discussion section.

2. There is a suggestion that loss of lin-2 and/or frm-3 unveils a trans-synaptic retrograde pathway that controls presynaptic release of acetylcholine. The authors mention such a pathway very briefly in the Introduction and the Results. After that, the authors devote a very large section of the Discussion to retrograde signaling, and they draw parallels to other modalities previously described.

Based on the data, this reviewer understands why the authors are inferring a retrograde pathway. However, it seems like there are other interpretations of the data. Those alternate possibilities should be considered and mentioned alongside the comments about retrograde signaling in all parts of the manuscript where it appears.

The authors infer that retrograde signaling is occurring because postsynaptic loss of either lin-2 or frm-3 results in decreased quantal content, as well as decreased spontaneous mini frequency. The implication is that the loss of postsynaptic scaffolds somehow changes presynaptic release probability of acetylcholine. This is quite possible. But it also possible that the decreased quantal content and mini frequency are entirely due to disrupted ACh receptor organization. This sort of result has been observed at many model synapses, including NMJs. Therefore, diminishment of neurotransmitter receptor clusters could result in observed decreases in quantal content as well as decreases in mini frequency.

Alternatively, could it be the case that the miniature amplitudes are so small in the mutants that some are not so easily discerned? This would be consistent with the decreased frequency. If the real mini amplitude were smaller in the mutants than reported, then the estimated quantal content might not actually be diminished, and the retrograde signaling model would not necessarily apply for that scenario.

Minor Points

1. There is a small question about the conclusions drawn from the deletion experiments for LIN-2: are the authors certain that the versions of LIN-2 lacking either the PDZ or SH3 domains were still functional? If these domain deletions were to disrupt protein structure, then that might create a null condition. The observed results would be the same, but the reason would not be domain-specific.

2. The authors should be congratulated for generating and characterizing new null alleles. This is not a trivial task, and it offers a chance to clear up prior studies (for example, where frm-3(gk585) was referred to as a null).

The new nulls are used effectively to argue (via phenotypic non-additivity) that FRM-3 and LIN-2 are operating in the same pathway. One question for this reviewer – was it surprising that the double hypomorphic conditions did not show additivity? Unlike the nulls, it should be possible to enhance phenotypes of hypomorphs.

Reviewer #3: CASK (encoded by lin2) is a MAGUK family scaffold protein with CaM, PDZ, SH3 and GK domains. FARP (encoded by frm3) is a protein containing FERM, RhoGEF, and PH domains. Both have been implicated in synapse formation. Li et al. focused on two mutants lin2(e1309) and frm3(gk585) of C. elegans. The lin-2(e1309) mutation deletes an N-terminal region of LIN-2A but has no effect on LIN-2B whereas the gk585 allele is a deletion disrupting both frm-3a and b. Phenotypes of these two mutations on cholinergic and GABAergic NMJs were described. First, they authors showed that the mutants displayed locomotor deficits. Second, both mEPSC amplitude and frequency were reduced. Double mutants showed similar defects to those in single mutants. mIPSC frequencies were unaltered in lin2(e1309) and frm3(gk585) mutants but were dramatically decreased in null mutants (lin2null or frm3null). The authors selectively deleted the genes in motor neurons or muscles, and observed impaired mEPSCs and evoked EPSCs only by muscle-specific deletion. They also observed reduced abundance of synaptic AChRs, but not overall expression in muscles, in null mutant animals. They further demonstrated that lin2 directly binds to the AChR through its SH3 domain, which requires the interaction between lin2 and frn3 at the PDZ-FERM domains. Because the frequency of EPSCs was reduced by muscle-specific mutation, but not motor neuron-selective mutation, the authors proposed a model of retrograde signaling by lin2 and frm3.

The paper provided some solid genetic evidence that lin2 and frm3 are involved in cholinergic synapses and GABAergic synapses. Presynaptic deficits in muscle-specific mutants seemed to be interesting, however, Zhou et al., JCB, 2021 had similar conclusion by a muscle-specific rescue. The Zhou 2021 paper also demonstrated the synergistic interaction of lin2 and frm3. Thus, the novelty here was questioned.

Second, the paper overall seemed to be descriptive, without much mechanistic insight on differential regulation on cholinergic versus GABA synapses and on molecular nature of retrograde signaling. The authors have much cleaner mutants (null and muscle- and motor neuron-specific cKOs), but did not carefully characterize those. The authors suggest that presynaptic and locomotion deficits were due to postsynaptic loss of lin2 and frm3, but clean muscle-specific mutants were not systematically for locomotion, AChR abundance, morphology of presynaptic terminals, and the number and priming of synaptic vesicles. The authors have the lines and tools and should provide such results.

Third, it was unclear why lin2(e1309) and frm3(gk585) were focused, which miss a particular domain or disrupt different isoforms. However, whether these domains/isoforms are involved were not certain. A reasonable experiment was to determine which phenotypes in muscle-specific cKO could be rescued by lin2(e1309) and frm3(gk585) mutants.

Fourth, the paper seemed to focus on “development” of synapses, but failed to present a time-course of different phenotypes. For example, which deficits (pre versus post-synaptic) occurs earlier, when?

Fifth, the authors stated that “Detection of spontaneous mEPSCs is impaired when their amplitudes are decreased; consequently, the decreased mEPSC frequency in lin-2(e1309) and frm-3(gk585) mutants could be caused by the smaller mEPSC amplitude.” By the authors’ argument, the frequency reduction must result from specific loss of low-amplitude mEPSCs. Was this the case? It would be important to determine whether frequency reduction did not result from a preferred reduction of low-amplitude mEPSCs.

Sixth, the roles of lin-2 and frm-3 in the GABAergic synapses are poorly understood. Apparently, the authors have a niche here to explore this deeply. Why e1309 and gk585 mutants altered cholinergic synapses, but had no effect on the frequency of mIPSCs of GABAergic synapses? Does this suggest that GABAergic synaptic defects are secondary? How about domains or isoforms that were specifically impacted by these mutations? Were there GABAergic phenotypes (such as IPSC) in lin2 and frm3 Neuron- and Muscle-Cre cKOs? A careful time-course analysis of the types of synapses and rescue, as described above, would be informative. Relatedly, were there any deficits at GABAergic synapses in delete-SH3 and/or delete-PDZ mutants?

Other concerns

More credits should be given to two papers by Zhou et al. in the introduction. The statement of “…, the functions of CASK and its binding proteins at cholinergic synapses have yet to be explored” seemed to be inappropriate.

Regarding retrograde mechanisms, the hypothesis of the neurexin-neuroligin pathway needs more discussion. Neuroligin may be involved in the recruitment of GABAAR but not that of AChR in C. elegans. In fact, syndecan (rather than neuroligin) was suggested to play the role in the cholinergic synapses (Zhou et al., J. Cell Biol. 2021).

Discussion could be improved by comparing worm NMJs with those in mammals, in particular for conserved proteins such as rapsyn.

Figure 5 panel A, the value of the y-axis is inconsistent with the scale bar, the unit of the y-axis should be micrometer square rather than millimeter square? In panel B, the quantification data showed a comparable mean value between the three groups; however, the representative images showed obviously higher numbers of SVs in the mutant groups than that in the control group.

The 1019 and 1036 strains were referred to as “null mutants” at some places. This inconsistency should be avoided, both in the Results and Figures.

Figure 2, panels A-D, for the reduction of amplitude of PSCs, representative traces showed much more dramatic changes than the quantitative data. Same issue in Figure 3, panels B-C and Figure 8B,C.

Figure 4, panels A,B, representative images showed much more dramatic reduction than the quantitative data.

Figure 6, panel E, “syb1018” should be “syb1019”.

A graphic elucidation for the interaction between lin2, frm3, and AChRs and the related domains should be helpful.

**Have all data underlying the figures and results presented in the manuscript been provided?**

Reviewer #1: Yes

Reviewer #2: Yes

Reviewer #3: Yes

PLOS authors have the option to publish the peer review history of their article (what does this mean?). If published, this will include your full peer review and any attached files.

Reviewer #1: **Yes: **Brock Grill

Reviewer #2: No

Reviewer #3: No

---

## [Decision Letter · Decision Letter 1]

10 Oct 2022

Dear Dr Hu,

We are pleased to inform you that your manuscript entitled "CASK and FARP  localize two classes of post-synaptic ACh receptors thereby promoting cholinergic transmission" has been editorially accepted for publication in PLOS Genetics. Congratulations!

As you will see, there are few minor edits suggested by Reviewer 1 that you may consider while preparing the final draft of your manuscript for the production team (the editorial team will not need re-evaluate). 

Yours sincerely,

Dion Kai Dickman, PhD

Guest Editor

PLOS Genetics

Gregory P. Copenhaver

Editor-in-Chief

PLOS Genetics

Comments from the reviewers (if applicable):

Reviewer's Responses to Questions

**Comments to the Authors:**

Reviewer #1: The authors have substantially improved their paper with extensive textual and experimental revisions. The reviewer’s concerns have been thoroughly addressed.

The reviewer also appreciates more careful attention to prior studies on FRM-3 and LIN-2 done in both C. elegans and other organisms. This study represents a substantial step forward in the reviewer’s opinion.

Removal of commentary on trans-synaptic signaling from post to presynaptic terminals is appreciated and substantially improved the accuracy of conclusions drawn.

Additional experimental revisions on GABAergic transmission are a welcome addition, and further bolster an interesting, rigorous study.

>>>

Minor suggested edits

1) Please note that the reviewer could not find language discussing and citing Figure S5. If this reviewer is correct, this should be updated.

2) Reviewer recommends that Figure 5B and 6B be annotated with “ns” labels for comparisons between mutants and wt since the major point here is that differences in synaptic vesicle numbers and markers are not significant.

3) Authors make the following statement: “The mIPSC amplitude in delta SH3 and delta PDZ mutants was decreased to a similar extent to that in lin-2 null mutants (Figure 8G, H).”

Should this be Figure 8F, G?

Reviewer #2: In this revision manuscript, Li et al. describe their work on C. elegans scaffolding proteins CASK and FARP. The revision rebuttal and paper content are responsive to the reviews. In response to some of this reviewer’s major comments, the authors have pulled back a main conclusion about retrograde signaling. They have also added additional muscle-specific Cre data, and they have rounded various other datasets in the manuscript (e.g., locomotion speed and electrophysiology).

As I stated in the first round of review, taking the data and results together, the authors have assembled an impressive and convincing dataset demonstrating that postsynaptic LIN-2 and FRM-3 are important for NMJ function. In this reviewer’s opinion, this paper should interest many readers of PLOS Genetics, including synaptic physiologists and neuroscientists interested in synapse development. The authors generated a lot of valuable reagents for the field, especially new null alleles which shed light upon the functions of LIN-2 and FRM-3.

Both of my prior major concerns have been addressed (1 – framing better what is known and not known about these scaffolds; 2 – retrograde pathway suggestion). My minor comments have also been addressed.

Additionally, I read through all the comments from Reviewers 1 and 3, and I think the authors were responsive, including adding new data using nulls (Rev 1), confirmation of the CRE reagent (Rev 1), contextualizing the new paper with the recent Zhou et al. studies (Rev 3), and additional behavior and electrophysiology (Rev 3).

Reviewer #3: The authors have done a reasonably good job in revising the manuscript by additional experiments and by clarification. The ms has been improved.

**Have all data underlying the figures and results presented in the manuscript been provided?**

Reviewer #1: Yes

Reviewer #2: Yes

Reviewer #3: Yes

PLOS authors have the option to publish the peer review history of their article (what does this mean?). If published, this will include your full peer review and any attached files.

Reviewer #1: No

Reviewer #2: No

Reviewer #3: No

**Data Deposition**

http://datadryad.org/submit?journalID=pgenetics&manu=PGENETICS-D-22-00491R1

**Press Queries**

---

## [Editor Report · Acceptance letter]

18 Oct 2022

PGENETICS-D-22-00491R1 

CASK and FARP  localize two classes of post-synaptic ACh receptors thereby promoting cholinergic transmission 

Dear Dr Hu, 

We are pleased to inform you that your manuscript entitled "CASK and FARP  localize two classes of post-synaptic ACh receptors thereby promoting cholinergic transmission" has been formally accepted for publication in PLOS Genetics! Your manuscript is now with our production department and you will be notified of the publication date in due course.

With kind regards,

Zsofia Freund

PLOS Genetics

On behalf of:
